# Entropic Causal Inference: Identifiability and Finite Sample Results

**Spencer Compton**
MIT
MIT-IBM Watson AI Lab
scompton@mit.edu

**Murat Kocaoglu**
MIT-IBM Watson AI Lab
IBM Research
murat@ibm.com

**Kristjan Greenewald**
MIT-IBM Watson AI Lab
IBM Research
kristjan.h.greenewald@ibm.com

**Dmitriy Katz**
MIT-IBM Watson AI Lab
IBM Research
dkatzrog@us.ibm.com

## Abstract

Entropic causal inference is a framework for inferring the causal direction between two categorical variables from observational data. The central assumption is that the amount of unobserved randomness in the system is not too large. This unobserved randomness is measured by the entropy of the exogenous variable in the underlying structural causal model, which governs the causal relation between the observed variables. [15] conjectured that the causal direction is identifiable when the entropy of the exogenous variable is not too large. In this paper, we prove a variant of their conjecture. Namely, we show that for almost all causal models where the exogenous variable has entropy that does not scale with the number of states of the observed variables, the causal direction is identifiable from observational data. We also consider the minimum entropy coupling-based algorithmic approach presented by [15], and for the first time demonstrate algorithmic identifiability guarantees using a finite number of samples. We conduct extensive experiments to evaluate the robustness of the method to relaxing some of the assumptions in our theory and demonstrate that both the constant-entropy exogenous variable and the no latent confounder assumptions can be relaxed in practice. We also empirically characterize the number of observational samples needed for causal identification. Finally, we apply the algorithm on Tübingen cause-effect pairs dataset.

## 1 Introduction

Understanding causal mechanisms is essential in many fields of science and engineering [26, 29]. Distinguishing causes from effects allows us to obtain a causal model of the environment, which is critical for informed policy decisions [21]. Causal inference has been recently utilized in several machine learning applications, e.g., to explain the decisions of a classifier [1], to design fair classifiers that mitigate dataset bias [14, 32] and to construct classifiers that generalize [28].

Consider a system that we observe through a set of random variables. For example, to monitor the state of a classroom, we might measure *temperature, humidity* and *atmospheric pressure* in the room. These measurements are random variables which come about due to the workings of the underlying system, the physical world. Changes in one are expected to cause changes in the other, e.g., decreasing the temperature might reduce the atmospheric pressure and increase humidity. As long as there are no feedback loops, we can represent the set of causal relations between these variables using a directed acyclic graph (DAG). This is called the *causal graph* of the system. Pearl and others showed that

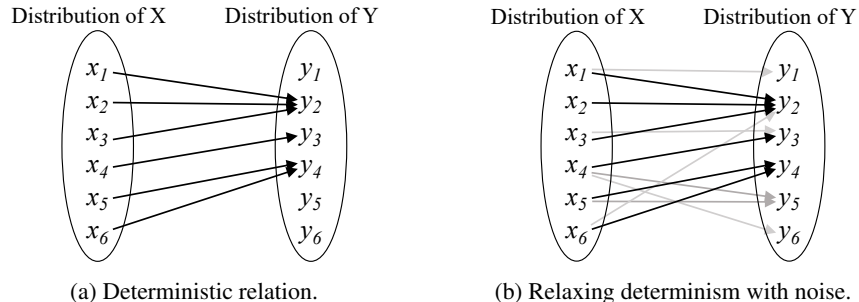

(a) Deterministic relation.          (b) Relaxing determinism with noise.

Figure 1: Intuition behind the entropic causality framework. **(a)** Most of the deterministic maps would be non-deterministic in the opposite direction, requiring non-zero additional randomness. **(b)** Entropic causality relaxes the deterministic map assumption to a map that needs low-entropy, and demonstrates that, most of the time, the reverse direction needs more entropy than the true direction.

knowing the causal graph enables us to answer many causal questions such as, *"What will happen if I increase the temperature of the room?"* [21].

Therefore, for causal inference, knowing the underlying causal structure is crucial. Even though the causal structure can be learned from experimental data, in many tasks in machine learning, we only have access to a dataset and do not have the means to perform these experiments. In this case, observational data can be used for learning some causal relations. There are several algorithms in the literature for this task, which can be roughly divided into three classes: Constraint-based methods and score-based methods use conditional independence statements and likelihood function, respectively, to output (a member of) the equivalence class. An equivalence class of causal graphs are those that cannot be distinguished by the given data. The third class of algorithms impose additional assumptions about the underlying system or about the relations between the observed variables. Most of the literature focus on the special case of two observed variables $X, Y$ and to understand whether $X$ causes $Y$ or $Y$ causes $X$ under different assumptions. Constraint or score-based methods cannot answer this question simply because observed data is not sufficient without further assumptions. In this work, we focus on the special case of two categorical variables. Even though the literature is more established in the ordinal setting, few results exist when the observed variables are categorical. The main reason is that, for categorical data, numerical values of variables do not carry any meaning; whereas in continuous data one can use assumptions such as smoothness or additivity [7].

We first start with a strong assumption. Suppose that the system is *deterministic*. This means that, even though observed variables contain randomness, the system has no additional randomness. When $X$ causes $Y$, this assumption implies that $Y = f(X)$ for some deterministic map $f(.)$. Consider the example in Figure 1. Since there is no additional randomness, each value of $X$ is mapped to a single value of $Y$. What happens if we did not know the causal direction and tried to fit a function in the wrong direction as $X = g(Y)$. Unlike $f$, $g$ has to be one-to-many: $Y = 2$ is mapped to three different value of $X$. Therefore, it is impossible to find a deterministic function in the wrong causal direction for this system. In fact, it is easy to show that most of the functions have this property: If $X, Y$ each has $n \geq 7$ states, all but $2^{-n}$ fraction of models can be identified.

Although there might be systems where determinism holds such as in traditional computer software, this assumption in general is too strict. Then *how much can we relax this assumption and still identify if $X$ causes $Y$ or $Y$ causes $X$?* In general, we can represent a system as $Y = f(X, E)$ where $E$ captures the additional randomness. To quantify this amount of relaxation, we use the entropy of the additional randomness in the structural equation, i.e., $H(E)$. For deterministic systems, $H(E) = 0$. This question was posed as a conjecture in [15], within the entropic causal inference framework.

We provide the first result in resolving this question. Specifically, we show that the causal direction is still identifiable for any $E$ with constant entropy. "Constant" is relative to the support size $n$ of the observed variables (note $0 \leq H(X) \leq \log(n)$). This establishes a version of Kocaoglu's conjecture.

A practical question is how much noise can the entropic causality framework handle: do we always need the additional randomness to not scale with $n$? Through experiments, we demonstrate that, in fact, we can relax this constraint much further. If $H(E) \approx \alpha \log(n)$, we show that in the wrong causal direction we need entropy of at least $\beta \log(n)$ for $\beta > \alpha$. This establishes that entropic causal

inference is robust to the entropy of noise and for most models, reverse direction will require larger entropy. We finally demonstrate our claims on the benchmark Tübingen dataset.

We also provide the first finite-sample analysis and provide bounds on the number of samples needed in practice. This requires showing finite sample bounds for the minimum entropy coupling problem, which might be of independent interest. The following is a summary of our contributions.

- We prove the first identifiability result for the entropic causal inference framework using Shannon entropy and show that for most models, the causal direction between two variables is identifiable, if the amount of exogenous randomness does not scale with $n$, where $n$ is the number of states of the observed variables.
- We obtain the first bounds on the number of samples needed to employ the entropic causal inference framework. For this, we provide the first sample-bounds for accurately solving the minimum entropy coupling problem in practice, which might be of independent interest.
- We show through synthetic experiments that our bounds are loose and entropic causal inference can be used even when the exogenous entropy scales with $\alpha \log(n)$ for $\alpha < 1$.
- We employ the framework on Tübingen data to establish its performance. We also conduct experiments to demonstrate robustness of the method to latent confounders, robustness to asymmetric support size, i.e., when $X, Y$ have very different number of states, and finally establish the number of samples needed in practice.

**Notation:** We will assume, without loss of generality, that if a variable has $n$ states, its domain is $[n] \coloneqq \{1, 2, \ldots, n\}$. $p(x)$ is short for $p(X = x)$. $p(Y|x)$ is short for the distribution of $Y$ given $X = x$. *Simplex* is short for probability simplex, which, in $n$ dimensions is the polytope defined as $\Delta_n \coloneqq \{(x_i)_{i \in [n]} : \sum_i x_i = 1, x_i \geq 0, \forall i \in [n]\}$. $\mathbb{1}_{\{\varepsilon\}}$ is the indicator variable for event $\varepsilon$. *SCM* is short for *structural causal model* and refers to the functional relations between variables. For two variables where $X$ causes $Y$, the SCM is $Y = f(X, E), X \perp\!\!\!\perp E$ for some variable $E$ and function $f$.

## 2 Identifiability with Entropic Causality

Consider the problem of identifying the causal graph between two observed categorical variables $X, Y$. We assume for simplicity that both have $n$ states, although this is not necessary for the results. Similar to the most of the literature, we make the causal sufficiency assumption, i.e., there are no latent confounders and also assume there is no selection bias. Then without loss of generality, if $X$ causes $Y$, there is a deterministic $f$ and an exogenous (unmeasured) variable $E$ that is independent from $X$ such that $Y = f(X, E)$, where $X \sim p(X)$ for some marginal distribution $p(X)$. Causal direction tells us that, if we intervene on $X$ and set $X = x$, we get $Y = f(x, E)$ whereas if we intervene on $Y$ and set $Y = y$, we still get $X \sim p(X)$ since $Y$ does not cause $X$.

Algorithms that identify causal direction from data introduce an assumption on the model and show that this assumption does not hold in the wrong causal direction in general. Hence, checking for this assumption enables them to identify the correct causal direction. Entropic causality [15] also follows this recipe. They assume that the entropy of the exogenous variable is bounded in the true causal direction. We first present their relevant conjecture, then modify and prove as a theorem.

**Conjecture 1** ([15]). *Consider the structural causal model $Y = f(X, E), X \in [n], Y \in [n], E \in [m]$ where $p(X), f, p(E)$ are sampled as follows: Let $p(X)$ be sampled uniformly randomly from the probability simplex in $n$ dimensions $\Delta_n$, and $p(E)$ be sampled uniformly randomly from the set of points in $\Delta_m$ that satisfy $H(E) \leq \log(n) + \mathcal{O}(1)$. Let $f$ be sampled uniformly randomly from all mappings $f : [n] \times [m] \to [n]$. Then with high probability, any $\tilde{E} \perp\!\!\!\perp Y$ that satisfies $X = g(Y, \tilde{E})$ for some mapping $g : [n] \times [m] \to [n]$ entails $H(X) + H(E) < H(Y) + H(\tilde{E})$.*

In words, the conjecture claims the following: Suppose $X$ causes $Y$ with the SCM $Y = f(X, E)$. Suppose the exogenous variable $E$ has entropy that is within an additive constant of $\log(n)$. Then, for most of such causal models, any SCM that generates the same joint distribution in the wrong causal direction, i.e., $Y$ causes $X$, requires a larger amount of randomness than the true model. The implication would be that if one can compute the smallest entropy SCM in both directions, then one can choose the direction that requires smaller entropy as the true causal direction.

We modify their conjecture in two primary ways. First, we assume that the exogenous variable has constant entropy, i.e., $H(E) = \mathcal{O}(1)$. Unlike the conjecture, our result holds for any such $E$. Second,

rather than the total entropy, we were able to prove identifiability by only comparing the entropies of the simplest exogenous variables in both directions $H(E)$ and $H(\tilde{E})$.[1] In Section 4, we demonstrate that both criteria give similar performance in practice.

Our technical result requires the following assumption on $p(X)$, which, for constant $\rho$ and $d$ guarantees that a meaningful subset of the support of $p(X)$ is *sufficiently uniform*. We will later show that this condition holds with high probability, if $p(X)$ is sampled uniformly randomly from the simplex.

**Assumption 1** (($\rho, d$)-uniformity). *Let $X$ be a discrete variable with support $[n]$. Then there exists a subset $S$ of size $|S| \geq dn$, such that $p(X = x) \in [\frac{1}{\sqrt{\rho}n}, \frac{\sqrt{\rho}}{n}], \forall x \in S$.*

We have the following theorem, which establishes that entropy in the wrong direction scales with $n$.

**Theorem 1** (Entropic Identifiability). *Consider the SCM $Y = f(X, E), X \perp\!\!\!\perp E$, where $X \in [n], Y \in [n], E \in [m]$. Suppose $E$ is any random variable with constant entropy, i.e., $H(E) = c = \mathcal{O}(1)$. Let $p(X)$ satisfy Assumption 1($\rho, d$) for some constants $\rho \geq 1, d > 0$. Let $f$ be sampled uniformly randomly from all mappings $f : [n] \times [m] \to [n]$. Then, with high probability, any $\tilde{E}$ that satisfies $X = g(Y, \tilde{E}), \tilde{E} \perp\!\!\!\perp Y$ for some $g$, entails $H(\tilde{E}) \geq (1 - o(1)) \log(\log(n))$. Specifically, for any $0 < r < q, H(\tilde{E}) \geq \left(1 - \frac{1+r}{1+q}\right) (0.5 \log(\log(n)) - \log(1 + r) - \mathcal{O}(1)), \forall n \geq \nu(r, q, \rho, c, d)$ for some $\nu$.*

Theorem 1 shows that when $H(E)$ is a constant, under certain conditions on $p(X)$, with high probability, the entropy of any causal model in the reverse direction will be at least $\Omega(\log(\log(n)))$. Specifically, if a constant fraction of the support of $p(X)$ contains probabilities that are not too far from $\frac{1}{n}$, our result holds. Note that *with high probability* statement is induced by the uniform measure on $f$, and it is relative to $n$. In other words, Theorem 1 states that the fraction of non-identifiable causal models goes to $0$ as the number of states of the observed variables goes to infinity. If a structure on the function is available in the form of a prior that is different than the uniform, this can potentially be incorporated in the analysis although we expect calculations to become more tedious.

Through the parameters $r, q$ we obtain a more explicit trade-off between the lower bound on entropy and how large $n$ should be for the result. $\nu(r, q, \rho, c, d)$ is proportional to $q$ and inversely proportional to $r$. The explicit form of $\nu$ is given in Proposition 1 in the supplement.

We next describe some settings where these conditions hold: We consider the cases when $p(X)$ has bounded element ratio, $p(X)$ is uniformly randomly sampled from the simplex, or $H(X)$ is large.

**Corollary 1.** *Consider the SCM in Theorem 1. Let $H(E) = c = \mathcal{O}(1)$ and $f$ be sampled uniformly randomly. Let $p(x)$ be such that either (a) $\frac{\max_x p(x)}{\min_x p(x)} \leq \rho$, or (b) $p(x)$ is sampled uniformly randomly from the simplex $\Delta_n$, or (c) $p(X)$ is such that $H(X) \geq \log(n) - a$ for some $a = \mathcal{O}(1)$.*

*Then, with high probability, any $\tilde{E}$ that satisfies $X = g(Y, \tilde{E}), \tilde{E} \perp\!\!\!\perp Y$ for some deterministic function $g$ entails $H(\tilde{E}) \geq 0.25 \log(\log(n)) - \mathcal{O}(1)$. Thus, there exists $n_0$ (a function of $\rho, c$) such that for all $n \geq n_0$, the causal direction is identifiable with high probability.*

The proof is given in Section G. Note we do not restrict the support size of the exogenous variable $E$.

**Proof Sketch of Theorem 1.** The full proof can be found in Appendix B.

1. Bound $H(\tilde{E})$ via $H(\tilde{E}) \geq H(X|Y = y), \forall y \in [n]$.
2. Characterize the sampling model of $f$ as a balls-and-bins game, where each realization of $Y$ corresponds to a particular bin, each combination $(X = i, E = k)$ corresponds to a ball.
3. Identify a subset of "good" bins $\mathcal{U} \subseteq [m]$. Roughly, a bin is "good" if it does not contain a large mass from the balls other than the ones in $\{(i, 1) : i \in S\}$.
4. Show one of the bins in $\mathcal{U}$, say $y = 2$, has many balls from $\{(i, 1) : i \in S\}$.
5. Bound the contribution of the most-probable state of $E$ to the distribution $p(X|Y = 2)$.
6. Characterize the effect of the other states of $E$ and identify a support for $X$ contained in $S$ on which the conditional entropy can be bounded. Use this to lower bound for $H(X|Y = 2)$.

**Conditional Entropy Criterion:** From the proof of Proposition 1 in Appendix B, we have $H(\tilde{E}) \geq \max_y H(X|Y=y) \geq (1 - o(1)) \log(\log(n))$. Further, we have $\max_x H(Y|X=x) \leq H(E) \leq c = \mathcal{O}(1)$. Hence not only is $H(\tilde{E}) > H(E)$ for large enough $n$, but $\max_y H(X|Y=y) > \max_x H(Y|X=x)$ as well. Therefore, under the assumptions of Theorem 1, $\max_y H(X|Y=y)$ and $\max_x H(Y|X=x)$ are sufficient to identify the causal direction:

**Corollary 2.** *Under the conditions of Theorem 1, we have that* $\max\limits_y H(X|Y=y) > \max\limits_x H(Y|X=x)$.

## 3 Entropic Causality with Finite Number of Samples

In the previous section, we provided identifiability results assuming that we have access to the joint probability distribution of the observed variables. In any practical problem, we can only access a set of samples from this joint distribution. If we assume we can get independent, identically distributed samples from $p(x, y)$, how many samples are sufficient for identifiability?

Given samples from $N$ i.i.d. random variables $\{(X_i, Y_i)\}_{i \in [N]}$ where $(X_i, Y_i) \sim p(x, y)$, consider the plug-in estimators $\hat{p}(y) := \frac{1}{\{N\}} \sum_{i=1}^N \mathbb{1}_{\{Y_i=y\}}$ and $\hat{p}(x, y) := \frac{1}{N} \sum_{i=1}^N \mathbb{1}_{\{X_i=x\}} \mathbb{1}_{\{Y_i=y\}}$ and define the estimator of the conditional $p(x|y)$ as $\hat{p}(x|y) := \frac{\hat{p}(x,y)}{\hat{p}(y)}$. Define $\hat{p}(x)$ and $\hat{p}(y|x)$ similarly.

**Definition 1.** *The minimum entropy coupling of $t$ random variables $U_1, U_2, \ldots, U_t$ is the joint distribution $p(u_1, \ldots, u_t)$ with minimum entropy that respects the marginal distributions of $U_i, \forall i$.*

The algorithmic approach of [15] relies on minimum entropy couplings. Specifically, they show the following equivalence: Given $p(x, y)$, let $E$ be the minimum entropy exogenous variable such that $E \perp\!\!\!\perp X$, and there exists an $f$ such that $Y = f(X, E), X \sim p(x)$ induces $p(x, y)$. Then the entropy of the minimum entropy coupling of the distributions $\{p(Y|x) : x \in [n]\}$ is equal to $H(E)$.

Therefore, understanding how having a finite number of samples affects the minimum entropy couplings allows us to understand how it affects the minimum entropy exogenous variable in either direction. Suppose $|\hat{p}(y|x) - p(y|x)| \leq \delta, \forall x, y$ and $|\hat{p}(x|y) - p(x|y)| \leq \delta, \forall x, y$. Given a coupling for distributions $p(Y|x)$, we construct a coupling for $\hat{p}(Y|x)$ whose entropy is not much larger. As far as we are aware, the minimum entropy coupling problem with sampling noise has not been studied.

Consider the minimum entropy coupling problem with $n$ marginals $\mathbf{p}_k = [p_k(i)]_{i \in [n]}, k \in [n]$. Let $p(i_1, i_2, \ldots, i_n)$ be a valid coupling, i.e., $\sum_{j \neq k} \sum_{i_j=1}^n p(i_1, i_2, \ldots, i_n) = p_k(i_k), \quad \forall k, i_k$. Consider the marginals with sampling noise shown as $\hat{\mathbf{p}}_k = [\hat{p}_k(i)]_{i \in [n]}, k \in [n]$. Suppose $|\hat{p}_k(i) - p_k(i)| \leq \delta, \forall i, k$. The following is shown in Section H of the supplement.

**Theorem 2.** *Let $p$ be a valid coupling for distributions $\{\mathbf{p}_i\}_{i \in [n]}$, where $\mathbf{p}_i \in \Delta_n, \forall i \in [n]$. Suppose $\{\mathbf{q}_i\}_{i \in [n]}$ are distributions such that $|\mathbf{q}_i(j) - \mathbf{p}_i(j)| \leq \delta, \forall i, j \in [n]$. If $\delta \leq \frac{1}{n^2 \log(n)}$, then there exists a valid coupling $q$ for the marginals $\{\mathbf{q}_i\}_{i \in [n]}$ such that $H(q) \leq H(p) + e^{-1} \log(e) + 2 + o(1)$.*

Theorem 2 shows that if the $l_\infty$ norm between the conditional distributions and their empirical estimators are bounded by $\delta \leq \frac{1}{n^2 \log(n)}$, there exists a coupling that is within 3 bits of the optimal coupling on true conditionals. To guarantee this with the plug-in estimators, we have the following:

**Lemma 1.** *Let $X \in [n], Y \in [n]$ be two random variables with joint distribution $p(x, y)$. Let $\alpha = \min\{\min_i p(X = i), \min_j p(Y = j)\}$. Given $N$ samples $\{(X_i, Y_i)\}_{i \in [N]}$ from independent identically distributed random variables $(X_i, Y_i) \sim p(x, y)$, let $\hat{p}(X|Y = y), \hat{p}(Y|X = x)$ be the plug-in estimators of the conditional distributions. If $N = \Omega(n^4 \alpha^{-2} \log^3(n))$, then $|\hat{p}(y|x) - p(y|x)| \leq \frac{1}{n^2 \log(n)}$ and $|\hat{p}(x|y) - p(x|y)| \leq \frac{1}{n^2 \log(n)}, \forall x, y$ with high probability.*

Next, we have our main identifiability result using finite number of samples:

**Theorem 3** (Finite sample identifiability). *Let $\mathcal{A}$ be an algorithm that outputs the entropy of the minimum entropy coupling. Consider the SCM in Theorem 1. Suppose $E$ is any random variable with constant entropy, i.e., $H(E) = c = \mathcal{O}(1)$. Let $p(X)$ satisfy Assumption 1($\rho, d$) for some constants $\rho \geq 1, d > 0$. Let $f$ be sampled uniformly randomly from all mappings $f : [n] \times [m] \to [n]$. Let $\alpha = \min\{\min_i p(X = i), \min_j p(Y = j)\}$. Given $N = \Omega(n^4 \alpha^{-2} \log^3(n))$ samples, let $\hat{p}(X|y), \hat{p}(Y|x)$ be the plug-in estimators for the conditional distributions. Then, for sufficiently large $n$, $\mathcal{A}(\{\hat{p}(X|y)\}_y) > \mathcal{A}(\{\hat{p}(Y|x)\}_x)$ with high probability.*

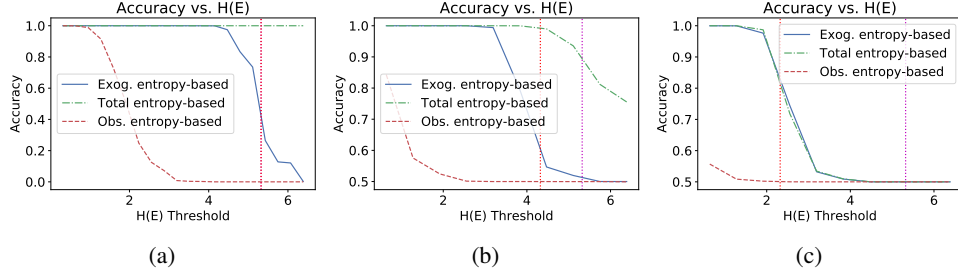

Figure 2: $m$ : number of states of $X$, $n$ : number of states of $Y$ in causal graph $X \rightarrow Y$. **(a)** $n = 40, m = 40$. Accuracy on simulated data: *Obs. entropy-based* declares $X \rightarrow Y$ if $H(X) > H(Y)$ and $Y \rightarrow X$ otherwise; *Exog. entropy-based* compares the exogenous entropies in both direction and declares $X \rightarrow Y$ if the exogenous entropy for this direction is smaller, and $Y \rightarrow X$ otherwise; *Total entropy-based* compares the total entropy of the model in both directions and declares the direction with smaller entropy as the true direction as proposed in [15]. **(b)** uses uniform mixture data from when $m = 40, n = 20$ and $m = 20, n = 40$. Similarly for **(c)** for $m = 40, n = 5$ and $m = 5, n = 40$. Magenta and red dashed vertical lines show $\log_2(\min\{m, n\})$ and $\log_2(\max\{m, n\})$, respectively.

From the equivalence between minimum entropy couplings and minimum exogenous entropy, Theorem 3 shows identifiability of the causal direction using minimum-entropy exogenous variables. Similar to Corollary 1, the result holds when $p(X)$ is chosen uniformly randomly from the simplex:

**Corollary 3.** *Consider the SCM in Theorem 1, where $H(E)=c=\mathcal{O}(1)$, $f$ is sampled uniformly randomly. Let $p(X)$ be sampled uniformly randomly from the simplex $\Delta_n$. Given $N=\Omega(n^8 \log^5(n))$ samples, let $\hat{p}(X|Y=y)$, $\hat{p}(Y|X=x)$ be the plug-in estimators for the conditional distributions. Then, for large enough $n$, $\mathcal{A}(\{\hat{p}(X|Y=y)\}_y) > \mathcal{A}(\{\hat{p}(Y|X=x)\}_x)$ with high probability.*

**Conditional Entropy Criterion with Finite Samples:** Note that the sample complexity in Theorem 3 scales with $\alpha^{-2}$ where $\alpha := \min\{\min_i p(X=i), \min_j p(Y=j)\}$. If either of the marginal distributions are not strictly positive, this can make the bound of Theorem 3 vacuous. To address this, we use an internal result from the proof of Theorem 1. In the proof we show that for some $i$, $p(Y=i) = \Omega(\frac{1}{n})$ and $H(X|Y=i) = \Omega(\log(\log(n)))$. Then, it is sufficient to obtain enough samples to accurately estimate $p(X|Y=i)$. Even though $i$ is not known a priori, since $p(Y=i) = \Omega(\frac{1}{n})$, estimating conditional entropies $H(X|Y=j)$ where the number of samples $|\{(x, Y=j)\}_x|$ exceeds a certain threshold guarantees that $p(X|Y=i)$ is estimated accurately. We have the following result:

**Theorem 4** (Finite sample identifiability via conditional entropy). *Consider the SCM in Theorem 1, where $H(E)=c=\mathcal{O}(1)$, $f$ is sampled uniformly randomly. Let $p(X)$ satisfy Assumption 1$(\rho, d)$ for some constants $\rho \geq 1, d > 0$. Given $N = \Omega(n^2 \log(n))$ samples, let $N_x$ be the number of samples where $X=x$ and similarly for $N_y$. Let $\hat{H}$ denote the entropy estimator of [30]. Then, for $n$ large enough, $\max_{\{y:N_y \geq n\}} \hat{H}(X|Y=y) > \max_{\{x:N_x \geq n\}} \hat{H}(Y|X=x)$ with high probability.*

Theorem 4 shows that $\mathcal{O}(n^2 \log(n))$ samples are sufficient to estimate the large conditional entropies of the form $H(Y|x), H(X|y)$, which in turn is sufficient for identifiability even for sparse $p(x, y)$.

## 4 Experiments

In this section, we conduct several experiments to evaluate the robustness of the framework. Complete details of each experiment are provided in the supplementary material. Unless otherwise stated, the greedy minimum entropy coupling algorithm of [15] is used to approximate $H(E)$ and $H(\tilde{E})$.

**Implications of Low-Exogenous Entropy Assumption.** We investigate the implications of this assumption. Specifically, one might ask if having low exogenous entropy implies $H(X) > H(Y)$. This would be unreasonable, since there is no reason for cause to always have the higher entropy.

In Figure 2, we evaluate the accuracy of the algorithm on synthetic data for different exogenous entropies $H(E)$. To understand the impact of the assumption on $H(X), H(Y)$, in addition to comparing exogenous entropies (*Exog. entropy-based*) and total entropies (*total entropy-based*)

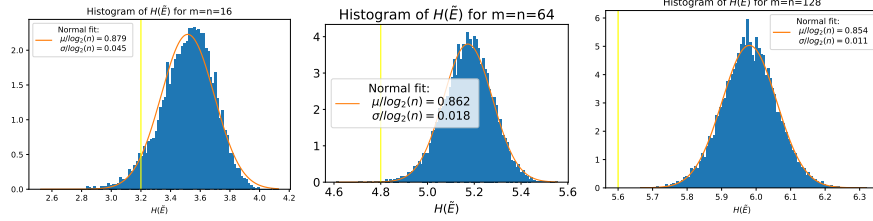

Figure 3: Histogram of $H(\tilde{E})$ when $H(E) \approx 0.8 \log_2(n)$. Yellow line shows $x = 0.8 \log_2(n)$

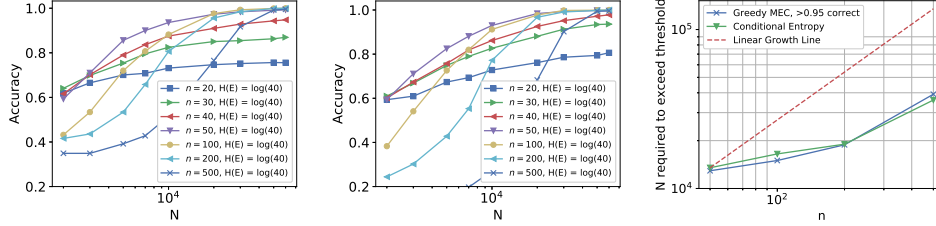

(a) Identification via conditional entropies ($H(E) \approx \log(40)$).

(b) Identification via MEC algorithm ($H(E) \approx \log(40)$).

(c) Number of samples vs. support size of observed variables.

Figure 4: (a) Probability of correctly discovering the causal direction $X \rightarrow Y$ as a function of $n$ and number of samples $N$, using the conditional entropies as the test. (b) Probability of correctly discovering the causal direction $X \rightarrow Y$ using the greedy MEC algorithm. (c) Samples $N$ required to reach 95% correct detection as a function of $n$, derived from the plots in Figure 4a and Figure 4b.

[15], we also show the performance of a simple baseline that compares $H(X)$ and $H(Y)$ (*obs. entropy-based*) and declares $X \rightarrow Y$ if $H(X) > H(Y)$ and vice versa.

We identify three different regimes, e.g., see Figure 2a: Regime 1: If $H(E) < 0.2 \log(n)$, we get $H(X) > H(Y)$ most of the time. All methods perform very well in this regime which we can call *almost deterministic*. Regime 2: If $0.2 \log(n) < H(E) < 0.6 \log(n)$, accuracy of *obs. entropy-based* method goes to 0 since, on average, we transition from the regime where $H(X) > H(Y)$ to $H(X) < H(Y)$. Regime 3: $0.6 \log(n) < H(E) < 0.8 \log(n)$ where $H(X) < H(Y)$ most of the time. As can be seen, *total entropy-based* and *exog. entropy-based* methods both show (almost) perfect accuracy in Regime $1, 2, 3$ whereas *obs. entropy-based* performs well only in Regime 1.

We also evaluated the effect of the observed variables having different number of states on mixture data in Figure 2b, 2c. In this case, framework performs well up until about $0.8 \log(\min\{m, n\})$.

**Relaxing Constant Exogenous-Entropy Assumption.** In Section 2, we demonstrated that the entropic causality framework can be used when the exogenous randomness is a constant, relative to the number of states $n$ of the observed variables. For very high dimensional variables, this might be a strict assumption. In this section, we conduct synthetic experiments to evaluate if entropic causality can be used when $H(E)$ scales with $n$. In particular, we test for various $\alpha < 1$ the following: *Is it true that the exogenous entropy in the wrong direction will always be larger, if the true exogenous entropy is $\leq \alpha \log(n)$?* For $\alpha = \{0.2, 0.5, 0.8\}$, we sampled 10k $p(E)$ from Dirichlet distribution such that $H(E) \approx \alpha \log(n)$ and calculated exogenous entropy in the wrong direction $H(\tilde{E})$. Figure 3 shows the histograms of $H(\tilde{E})$ for $\alpha = 0.8$ and $n = \{16, 64, 128\}$. We observe that $H(\tilde{E})$ tightly concentrates around $\beta \log(n)$ for some $\beta > \alpha$. For reference, $\alpha \log(n)$ is shown by the vertical yellow line. Similar results are observed for other $\alpha$ values which are provided in the supplementary material.

**Effect of Finite Number of Samples.** In Section 3, we identified finite sample bounds for entropic causality framework, both using the exogenous entropies $H(E), H(\tilde{E})$ and using conditional entropies of the form $\max_y H(X|Y=y), \max_x H(Y|X=x)$. We now test if the bounds are tight.

We observe two phases and a transition phenomenon in between. The first phase occurs for small values of $n$, for $n \in \{20, 30, 40\}$. Here, the fraction of identifiable causal models does not reach 1 as

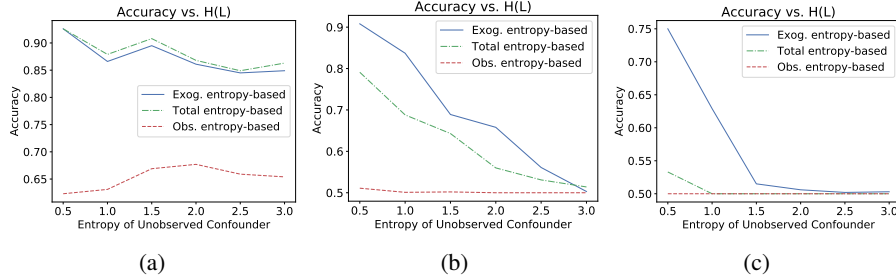

Figure 5: Accuracy on simulated data with *light* confounding. Number of states and data are identical to those in Figure 2. We use exogenous entropy of 2 bits and add a confounder $L$. This can be interpreted as replacing some bits of the exogenous variable in Figure 2 with those of a latent confounder. Surprisingly, performance for $H(E)=2, H(L)=t$ is similar to the performance when $H(E)=2+t$ in Figure 2. This indicates that the proposed method is robust to latent confounders, as long as the total exogenous and confounder entropy is not very close to $\min\{\log(n), \log(m)\}$.

| | | | | | | | |
|---|---|---|---|---|---|---|---|
| 5-state quantization | Threshold ($\times$ log support) | 0.7 | 0.8 | 0.85 | 0.9 | 1.0 | 1.2 |
| | # of pairs | 14 | 25 | 34 | 42 | 57 | 85 |
| | Accuracy (%) | 85.7 | 64.0 | 58.8 | 57.1 | 63.2 | 60.0 |
| **10-state quantization** | Threshold ($\times$ log support) | 0.7 | 0.8 | 0.85 | 0.9 | 1.0 | 1.2 |
| | # of pairs | 13 | 23 | 34 | 46 | 67 | 85 |
| | Accuracy (%) | 84.6 | 73.9 | 70.6 | 63.0 | 61.2 | 56.5 |
| 20-state quantization | Threshold ($\times$ log support) | 0.7 | 0.8 | 0.85 | 0.9 | 1.0 | 1.2 |
| | # of pairs | 12 | 21 | 41 | 52 | 76 | 85 |
| | Accuracy (%) | 75.0 | 61.9 | 53.7 | 51.9 | 51.3 | 49.4 |

Table 1: Performance on Tübingen causal pairs with low exogenous entropy in at least one direction.

the number of samples is increased, but saturates at a smaller value. This is expected since exogenous noise is relatively high, i.e., $H(E) \geq \log(n)$. For $n > 40$, or equivalently, when $H(E) \leq \log(n)$, increasing number of samples increases accuracy to 1, as expected.

The greedy MEC criterion has slightly better performance (by $\approx 5\%$), indicating more robustness. This may be due to a gap between $H(\tilde{E})$ and $H(X|Y=y)$ since greedy-MEC output is not limited by $\log(n)$ unlike conditional entropy. In contrast to the $\tilde{O}(n^8)$ bound, the number of samples needed has a much better dependence on $n$. Figure 4c includes a dashed linear growth line for comparison.

**Effect of Confounding** The equivalence between finding the minimum entropy exogenous variable and finding the minimum entropy coupling relies on the assumption that there are no unobserved confounders in the system. Despite lack of theory, it is useful to experimentally understand if the method is robust to *light confounding*. One way to assess the effect of confounding is through its entropy: If a latent confounder $L$ is a constant, i.e., it has zero entropy, it does not affect the observed variables. In this section, we simulate a system with light confounding by limiting the entropy of the latent confounder and observe how quickly it degrades the performance of entropic causality.

The results are given in Figure 5. The setting is similar to that of Figure 2. We set $H(E) \approx 2$ and show accuracy of the method as entropy of the latent $L$ is increased. Perhaps surprisingly, the effect of increasing the entropy of the confounder is very similar to the effect of increasing the entropy of the exogenous variable. This shows that the method is robust to light latent confounding.

**Tübingen Cause-Effect Pairs** In [15], authors employed the total entropy-based algorithm on Tübingen data [20] and showed that it performs similar to additive noise models with an accuracy of $64\%$. Next, we test if entropic causality can be used when we only compare exogenous entropies.

The challenge of applying entropic causality on Tübingen data is that most of the variables are continuous. Therefore, before applying the framework, one needs to quantize the data. The authors chose a uniform quantization, requiring both variables have the same number of states. We follow a similar approach. For $b \in \{5, 10, 20\}$, the value of $n$ is chosen for both $X, Y$ as the minimum of $b$, $N/10$, $N_x^{uniq}$ and $N_y^{uniq}$, where $N$ is the number of samples available for pair $X, Y$ and $N_x^{uniq}, N_y^{uniq}$ are the number of unique realizations of $X, Y$, respectively.

As a practical check for the validity of our key assumption, we make a decision based on the following: For a threshold $t$, algorithm makes a decision only for pairs for which either $H(E) \leq t \log(n)$ or $H(\tilde{E}) \leq t \log(n)$. We report the accuracies in Table 1. As we expect, for stricter thresholds, accuracy is improved, supporting the assumption that in real data, the direction with the smaller exogenous entropy is likely to be the true direction. The most consistent performance is obtained for $b = 10$.

To check the stability of performance to quantization, we conducted an experiment where we perturb the quantization intervals and take majority of 5 independent decisions. This achieves qualitatively similar (sometimes better, sometimes worse) performance shown in Table 3 in the appendix. Exploring best practices to quantize continuous data is an interesting avenue for future work.

We now compare performance with other leading methods on this dataset. The total-entropy approach for Entropic Causal Inference achieved $64.21\%$ accuracy at $100\%$ decision rate in [15]. ANM methods are evaluated on this data in [20], where they emphasize two ANM methods with consistent performance that achieve $63 \pm 10\%$ and $69 \pm 10\%$ accuracy. IGCI methods are also evaluated in [20] and were found to vary greatly with implementation and perturbations of data. No IGCI method had consistent performance. LiNGAM methods are evaluated in [8] and reported nonlinear approaches with $62\%$ and $69\%$ accuracy. Of these, only Entropic Causal Inference and IGCI can handle categorical data. Comparison is difficult with limited data, but we suggest assessing the MEC in both directions when deciding how to use our approach in combination with other methods.

## 5    Discussion

In this section we discuss several aspects of our method in relation with prior work. First, note that our identifiability result holds *with high probability* under the measure induced by our generative model. This means that, even under our assumptions, not all causal models will be identifiable. However, the non-identifiable fraction vanishes as $n$, i.e., the number of states of $X, Y$ goes to infinity. In essence, this is similar to many of the existing identifiability statements that show identifiability except for an adversarial set of models [7]. Specifically in [15], the authors show that under the assumption that the exogenous variable has small support size, causal direction is identifiable with probability 1. This means that the set of non-identifiable models has Lebesgue measure zero. This is clearly a stronger identifiability statement. However, this is not surprising if we compare the assumptions: Bounding the support size of a variable bounds its entropy, but not vice verse. Therefore, our assumption can be seen as a relaxation of the assumption of [15]. Accordingly, a weaker identifiability result is expected.

Next, we emphasize that our key assumption, that in the true causal direction the exogenous variable has small entropy, is not universal, i.e., one can construct cause-effect pairs where the anti-causal direction requires less entropy. [9] provides an example scenario: Consider a ball traveling at a fixed and known velocity from the initial position $X$ towards a wall that may appear or disappear at a known position with some probability. Let $Y$ be the position of the ball after a fixed amount of time. Clearly we have $X \rightarrow Y$. If the wall appears, the ball ends up in a different position ($y_0$) from the one it would if the wall does not ($y_1$). Then the mapping $X \rightarrow Y$ requires an exogenous variable to describe the behavior of the wall. However, simply by looking at the final position, we can infer whether wall was active or not, and accordingly the initial position deterministically. This shows that our key assumption is not always valid and should be evaluated depending on the application in mind.

Finally note that the low-entropy assumption should not be enforced on the exogenous variable of the cause, since this would imply that $X$ has small entropy. This brings about a conceptual issue to extend the idea to more than two variables: Which variables' exogenous noise should have small entropy? For that setting, we believe the original assumption of [15] may be more suitable: Assume that the total entropy of the system is small. In the case of more than two variables, this means total entropy of all the exogenous variables is small, without enforcing bounds on specific ones.

## 6    Conclusion

In this work, we showed the first identifiability result for learning the causal graph between two categorical variables using the entropic causal inference framework. We also provided the first finite-sample analysis. We conducted extensive experiments to conclude that the framework, in practice, is robust to some of the assumptions required by theory, such as the amount of exogenous entropy and causal sufficiency assumptions. We evaluated the performance of the method on Tübingen dataset.

## Acknowledgments and Disclosure of Funding

This work was supported by the MIT-IBM Watson AI Lab.

## Broader Impact

Determining causal direction from data has numerous applications. The main challenge in using purely observational data for causal inference always lies in the set of assumptions that are made. Especially for safety-critical applications, the assumptions should be very carefully evaluated.

In this work, we use the assumption that the exogenous variables have small entropy. This means that the factors which affect the effect variable have only a small number of states that are active, relative to the number of active states of the cause and effect variables. Only then there is some structure in the probability distribution in the context of entropic causality. Otherwise, the structure disappears and the approach will be unreliable. In an application, this assumption should first be evaluated by the field experts. If the expert believes this assumption might be violated, other observational methods that rely on a different set of assumptions should be used instead.

## Footnotes

[1] Entropy of the exogenous variable, or in the case of Conjecture 1 the entropy of the system, can be seen as a way to model complexity and the method can be seen as an application of Occam's razor. In certain situations, especially for ordinal variables, it might be suitable to also consider the complexity of the functions.

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
