[Supplementary Material]

# Supplementary Material
# Entropic Causal Inference: Identifiability and Finite Sample Results

## A    Related Work

There are a variety of assumptions and accompanying methods for inferring the causal relations between two observed variables [27, 22, 18, 4, 5]. For example, authors in [7] developed a framework to infer causal relations between two continuous variables if the exogenous variables affect the observed variable additively. This is called the *additive noise model (ANM)*. Under the assumption that the functional relation is non-linear they show identifiability results, i.e., for almost all models the causal direction between two observed variables can be identified. This is typically done by testing independence of the residual error terms from the regression variables. Interestingly in [17] authors show that independence of regression residuals leads the total entropy in the true direction to be smaller than the wrong direction, which can be used for identifiability thereby arriving at the same idea we use in our paper.

A challenging setting for causal inference is the setting with discrete and categorical variables, where the variable labels do not carry any specific meaning. For example, *Occupation* can be mapped to discrete values $\{0, 1, 2, \ldots\}$ as well as to one-hot encoded vectors. This renders methods which heavily rely on the variable values, such as ANMs, unusable. While extensions of ANMs to the discrete setting exist, they still utilize the variable values and are not robust to permuting the labels of the variables. One related approach proposed in [11] is motivated by Occam's razor and proposes to use the Kolmogorov complexity to capture the complexity of the causal model, and assume that the true direction is "simple". As Kolmogorov complexity is not computable, the authors resort to a proxy, based on minimum description length.

Another line of work uses the idea that causes are *independent* from the causal mechanisms, which is called the *independence of cause and mechanism* assumption. The notion of independence should be formalized since comparison is between a random variable and a functional relation. In [10, 12], authors propose using information geometry within this framework to infer the causal direction in deterministic systems. Specifically, they create a random variable using the functional relation based on uniform distribution and utilize the hypothesis that this variable should be independent from the cause distribution.

## B    Proof of Theorem 1

**Step 1. Bounding $H(\tilde{E})$ by $H(\tilde{E}) \geq H(X|Y = y), \forall y$:** Consider any $\tilde{E} \perp\!\!\!\perp Y$ for which there exists a deterministic map $g$ such that $X = g(\tilde{E}, Y)$. We have

$$p(X = x|Y = y) = p(g(\tilde{E}, Y) = x|Y = y)$$
$$= p(g(\tilde{E}, y) = x) = p(g_y(\tilde{E}) = x),$$

for $g_y(e) := g(e, y), \forall e, y$, since $\tilde{E} \perp\!\!\!\perp Y$. Due to data processing inequality, it follows that $H(\tilde{E}) \geq H(X|Y = y)$.

In [15], this analysis is used to show that the minimum entropy exogenous variable $\tilde{E}$ can be obtained by solving the minimum entropy coupling problem on the conditional distributions $p(X|Y = y)$. Here, we use the conditional entropies to lower bound the entropy of the exogenous variable $\tilde{E}$. Therefore, in the rest of our analysis we attempt to show that under the given assumptions, with high probability, $H(X|Y = y)$ is large for some value of $y$.

**Step 2. Generative process as a balls and bins game:** In order to analyze the conditional distributions $p(X|Y = y)$ we relate the generative model to a balls and bins game:

Consider a deterministic map $f : [n] \times [m] \to [n]$. Let $p(X = i) = x_i$ and $p(E = k) = e_k$. Without loss of generality, assume that $X$ and $E$ are labeled in decreasing probability order. In other words, $e_k \geq e_l$ if $k < l$ and $x_i \geq x_j$ if $i < j$.[2] Let $\mathbf{M}$ be the matrix defined as $\mathbf{M}_{i,k} := f(i, k)$. The

probability distribution $p(Y|X)$ is determined by the causal mechanism, i.e., the structural equation $Y = f(X, E)$. The conditional distributions in the wrong causal direction, i.e., $p(X|Y)$ can then be calculated as follows:

$$p(X = i|Y = j) = \frac{1}{Z}x_i \sum_{k=1}^{m} \mathbb{1}_{\{\mathbf{M}_{i,k}=j\}}e_k.$$

$Z = \sum_{i=1}^{n} x_i \sum_{k=1}^{m} \mathbb{1}_{\{\mathbf{M}_{i,k}=j\}}e_k$ is the normalizing constant.

To sample $f$ uniformly randomly from all the mappings is equivalent to filling the entries of $\mathbf{M}$ independently and uniformly randomly from $\mathcal{Y} = [n]$. A small example is given in Table 1, which shows a realization of $f$ through matrix $\mathbf{M}$, and illustrates how this affects $p(X|Y = 1)$.

| | $\mathcal{E}$ | 1 | 2 | 3 | 4 | 5 |
|---|---|---|---|---|---|---|
| $\mathcal{X}$ | PMF of $E$ / PMF of $X$ | $e_1$ | $e_2$ | $e_3$ | $e_4$ | $e_5$ |
| 1 | $x_1$ | 2 | 3 | 2 | 1 | 1 |
| 2 | $x_2$ | 3 | 2 | 3 | 3 | 1 |
| 3 | $x_3$ | 3 | 1 | 2 | 3 | 2 |

| | $\mathbb{P}(X = x|Y = 1)$ |
|---|---|
| $x = 1$ | $\frac{x_1(e_4+e_5)}{Z}$ |
| $x = 2$ | $\frac{x_2 e_5}{Z}$ |
| $x = 3$ | $\frac{x_3 e_2}{Z}$ |

Table 2: Left: Balls and bins representation of function $f : \mathcal{X} \times \mathcal{E} \to \mathcal{Y}$, where $\mathcal{X} = \mathcal{Y} = [3]$ and $\mathcal{E} = [5]$. The function values for a given $X = i, E = k$ can be seen as realizations of a two dimensional balls and bins game. Right: Conditional probability values of $X$ given $Y = 1$ for the given function. $Z = x_1(e_1 + e_3) + x_2(e_2) + x_3(e_5)$ is the normalization constant, which also gives $\mathbb{P}(Y = 1)$.

Any realization of $f$ corresponds to a realization of matrix $\mathbf{M}$. The first column is of special interest to us because it corresponds to the value of $E$ with the highest probability. The realization of $\mathbf{M}$ can be thought of as a balls and bins process, with the cells corresponding to balls and each entry $\mathbf{M}_{i,k}$ corresponding to which bin that cell's ball landed in.

**Step 3. Identify a set of "good" bins:** Each coordinate $(i, k)$ is a ball and the value of $\mathbf{M}_{i,k}$ is the identity of the bin this ball is placed in. We utilize the existence of a set $S$ as described in the theorem statement as follows: We focus on the set of balls corresponding to the cells $(i, 1)$ for $i \in S$. Our goal is to identify a bin which contains a large fraction of these balls. We also want this bin to not contain too much probability mass from balls outside of the set $S$ in order to get a close bound in **Step 6**.

Recall that each bin $y$ contains mass $x_i e_k$ when $\mathbf{M}_{i,k} = y$. To restrict our search of a good bin, we first discard all the bins that contain a large mass from entries of $\mathbf{M}$ that are either in rows corresponding to $x \notin S$ or columns other than the first column. Let $p(X, Y, E)$ represent the joint distribution between $X, Y, E$. Then we discard every value of $y$ where $\sum_{x \notin S} \sum_{e=1}^{m} p(x, y, e) + \sum_{x \in S} \sum_{e=2}^{m} p(x, y, e)$ is large. We pick the threshold of $\frac{2}{n}$ and define the set $\mathcal{B}$ accordingly:

$$\mathcal{B} = \left\{y : \sum_{x \notin S} p(x, y) + \sum_{x \in S} p(x, y, E > 1) > \frac{2}{n}\right\}.$$

We know that $|\mathcal{B}| \leq \frac{n}{2}$, since otherwise the total mass would exceed $1$.[3] Let $\mathcal{U} := [n] \backslash \mathcal{B}$. Then $|\mathcal{U}| \geq n/2$. Note that $\mathcal{B}$ and $\mathcal{U}$ are determined in a manner not affected by the realized values of $\mathbf{M}_{x,1}$ for $x \in S$. We will next focus on only the values of $y \in \mathcal{U}$, and later quantify the following claim: A significant fraction of the probability mass that falls in any bin in $\mathcal{U}$ is due to entries from $\mathbf{M}_{x,1}$ for $x \in S$. Therefore, for one of these bins $y \in \mathcal{U}$, we can focus on obtaining a lower bound of $H(X|Y = y, X \in S, \mathbf{M}_{X,1} = y)$ to later show that $H(X|Y = y)$ cannot be much smaller.

**Step 4. Show a bin from $\mathcal{U}$ has many balls from the first column of $\mathbf{M}$ and rows in $S$:** We focus our attention to the balls in $S$ and bins in $\mathcal{U}$. We want to show that $\exists y \in \mathcal{U}$ such that $\mathbf{M}_{x,1} = y$ for a large number of values of $x \in S$. Recall that since $|S| \geq dn$, we have at least $dn$ balls falling into $n$

bins. Moreover, since $|\mathcal{U}| \geq n/2$, at least $n/2$ of these bins are "good" for us. First, we show that, with high probability, at least $\frac{dn}{4}$ of the $dn$ balls fall in the bins in $\mathcal{U}$.

**Lemma 2.** *Consider the process of uniformly randomly throwing $dn = \Theta(n)$ balls into $n$ bins.[4] Let $\mathcal{U}$ be an arbitrary, fixed subset of bins with size $|\mathcal{U}| \geq \frac{n}{2}$. Then with high probability, at least $\frac{dn}{4}$ balls fall into the bins in $\mathcal{U}$. Moreover, these balls are also uniformly randomly thrown.*

The above lemma, proven in Appendix C is directly applicable to our setting, even though $\mathcal{U}$ is a random variable. This is because the realization of the entries of $\mathbf{M}$ outside the rows $S$ or outside the first column, which determines the set $\mathcal{U}$ are independent from the entries in $\mathbf{M}$ in the rows $S$ and in the first column. In other words, how balls are thrown into the bins in $\mathcal{U}$ is not affected by how $\mathcal{U}$ is chosen.

We want to use this to show that there is a bin $y \in \mathcal{U}$ such that the conditional distribution $p(X|Y = y)$ is due to many balls $x \in S$ where $\mathbf{M}_{x,1} = y$. We have shown that with high probability at least $\frac{dn}{4}$ balls land in bins corresponding to $y \in \mathcal{U}$. We apply a bound from Theorem 1 of [24], which implies that with high probability when there are $b$ bins and $\eta b$ balls ($\eta = \Theta(1)$), the most loaded bin has at least $\frac{\ln(b)}{\ln(\ln(b)) + \ln(\frac{1}{\eta})}$ balls. We know that with high probability we have some number of balls in range $[\frac{nd}{4}, nd]$ in some number of good bins in range $[\frac{n}{2}, n]$. In terms of the established bound on the most loaded bin, this means $\eta \geq \frac{d}{4}$ and $b \in [\frac{n}{2}, n]$. If we substitute valid values of $\eta$ and $b$ that minimize the lower bound, we know that with high probability the heaviest loaded bin among $\mathcal{U}$ conditional distributions has at least $\frac{\ln(n) - \ln(2)}{\ln(\ln(n)) + \ln(\frac{4}{d})}$ balls. Without loss of generality, suppose this bin has label 2. We show that $H(X|Y = 2)$ is large using the above bound.

**Step 5. Bounding $H(X|Y = 2)$:** Next, we obtain a lower bound for $H(X|Y = 2)$. We utilize the following lemma, proved in Section D of the supplement:

**Lemma 3.** *Let $X$ be a discrete random variable with distribution $[p_1, p_2, \ldots, p_n]$. Consider the random variable $X'$ with distribution $[\frac{p_i}{\sum_{j \in S'} p_j}]_i$ for any $S' \subseteq [n]$. Then, $H(X) \geq \mu H(X')$, where $\mu = \sum_{i \in S'} p_i$.*

To use this lemma, we consider a specific distribution induced on the support of $X|Y = 2$. First, let us define the following: For any subset $S' \subseteq [n], y \in [n]$, let $X_{S',y}$ be the discrete variable with the following distribution:

$$p(X_{S',y} = i) = \frac{p(X = i|Y = y)}{\sum_{l \in S'} p(X = l|Y = y)}, \forall i \in S'. \tag{1}$$

We focus on $X_{S',2}$, where $S' = \{i : i \in S, \mathbf{M}_{i,1} = 2\}$. We first show that $H(X_{S',2})$ is large, and then show the total mass $\mu = \sum_{i \in S'} p(X = i|Y = 2)$ that $X_{S',2}$ contributes to $(X|Y = 2)$ is large, which allows us to use Lemma 3.

To show $H(X_{S',2})$ is large, we use the following lemma from [3]:

**Lemma 4** (Theorem 2 of [3]). *Let $X$ be a strictly positive discrete random variable on $n$ states such that $\frac{\max_i p(X=i)}{\min_i p(X=i)} \leq \rho$. Then*

$$H(X) \geq \log(n) - \left(\frac{\rho \ln(\rho)}{\rho - 1} - 1 - \ln\left(\frac{\rho \ln(\rho)}{\rho - 1}\right)\right) \frac{1}{\ln(2)}.$$

To lower bound $H(X_{S',y})$ using the above lemma, we obtain an upper bound to $\rho' := \frac{\max_i p(X_{S',2} = i)}{\min_i p(X_{S',2} = i)}$ by utilizing our knowledge that $H(E) = c$. For each value $i \in S'$, we know that $\mathbf{M}_{i,1} = 2$. Thus, $p(X_{S',2} = i) \geq \frac{x_i e_1}{\mu}$. Also $p(X_{S',2} = i) \leq \frac{x_i \sum_{k=1}^{m} e_k}{\mu} = \frac{x_i}{\mu}$ and $\frac{\max_{i \in S} x_i}{\min_{i \in S} x_i} \leq \rho$. Therefore $\rho' \leq \frac{\max_i \frac{x_i}{\mu}}{\min_i \frac{x_i e_1}{\mu}} \leq \frac{\rho}{e_1}$.

In order to understand how small $e_1$ can be under the given constraints, we obtain a useful characterization for constant entropy distributions. The following lemma shows that the maximum probability value for any discrete distribution with constant entropy is a constant away from zero.

**Lemma 5.** *Let $E$ be a discrete random variable with $m$ states, with the probability distribution $[e_1, e_2, \ldots, e_m]$, where without loss of generality $e_i \geq e_j, \forall j > i$. If $H(E) \leq c$ then $e_1 \geq 2^{-c}$.*

The proof is given in Section E in the supplement.

Applying Lemmas 3-5, with some derivation we show in Section F of the supplement that:

**Proposition 1** (**Step 6**). *Under the conditions stated above,*

$$H(\tilde{E}) \geq \max_y H(X|Y = y) \geq H(X|Y = 2)$$
$$\geq (1 - o(1))[\log(\log(n)) - \log(\log(\log(n))) - \mathcal{O}(1)].$$

*Furthermore, to make the trade-off between the strength of the lower bound and assumptions on $n$ more explicit, when $n \geq \nu(r, q, \rho, c, d)$ with*

$$\nu(r, q, \rho, c, d) = \max\{4, e^{\left(\frac{4}{d}\right)^{1/r}}, 2e^{q^2 2^{2(c+1)}\rho}\},$$

*we have*

$$H(\tilde{E}) \geq \max_y H(X|Y = y) \geq H(X|Y = 2)$$
$$\geq \left(1 - \frac{1+r}{1+q}\right)(0.5\log(\log(n)) - \log(1 + r) - \mathcal{O}(1)).$$

This completes the proof of Theorem 1. $\qquad\square$

**Potential Improvements and Limitations:** In our analysis, we use $\max_y H(X|Y = y)$ to bound $H(\tilde{E})$. One potential improvement might be obtained by considering the gap between $H(\tilde{E})$ and the collection $\{H(X|Y = y)\}_y$ for a given $p(x, y)$. [15] showed that the smallest $H(\tilde{E})$ is given by the minimum entropy coupling of the conditional distributions $\{p(X|Y = y)\}_y$. Follow-up works have developed minimum-entropy coupling algorithms [2, 16, 25] and obtained approximation guarantees. However there is currently no tight analysis characterizing this entropy gap.

Note that the original conjecture proposes that $H(E) \leq \log(n) + \mathcal{O}(1)$ is sufficient. This is a very strong statement and we believe, even if it is true, it requires a much deeper understanding on the minimum entropy couplings than is currently available in the literature. We do, however, provide evidence in Section 4 that $H(E) \leq \alpha \log(n)$ for $\alpha < 1$ seems sufficient for identifiability.

One point in our analysis that is related to this setting when $H(E)$ scales with $n$, is that we only considered the first column of the matrix $\mathbf{M}$, i.e., we have only taken into account the probability values of the form $x_i e_1$ contributing to the entropy of $H(X|Y = y)$. As long as the function $f$ is sampled uniformly randomly in the considered generative model, this approach cannot give $H(\tilde{E}) \gg \log(\log(n))$ due to the support size of $X$ being upper bounded by $\mathcal{O}(\log(n))$ with high probability from the balls and bins perspective. For when $H(E)$ is very small, we do expect this to be a reasonable approach as the remaining columns have very small probability values, hence very small impact. However, for going beyond the current analysis and for proving identifiability when $H(E)$ scales with $n$, we strongly believe that the effect of the remaining columns should be considered.

## C  Proof of Lemma 2

Let $\varepsilon$ be the event that less than $\frac{dn}{4}$ balls fall in the bins in $\mathcal{U}$. We provide an upper bound for the probability of this event $P(\varepsilon)$. Consider the indicator variables each corresponding to the event that a particular ball lands in $\mathcal{U}$. These indicator variables are independently and identically distributed, where each has probability $\frac{\mathcal{U}}{n} \geq \frac{1}{2}$ of being 1. We use Hoeffding's inequality to bound $P(\varepsilon)$. Let $S_{dn}$ be the sum of the $dn$ indicator variables (i.e., the number of the balls that land in bins corresponding

to $\mathcal{U}$) and $E_{dn}$ be the expected sum of the indicator variables ($E_{dn} = dn\left(\frac{\mathcal{U}}{n}\right)$).

$$P(\varepsilon) = P\left(S_{dn} < \frac{dn}{4}\right) \tag{2}$$

$$\leq P\left(|S_{dn} - E_{dn}| > \left|E_{dn} - \frac{dn}{4}\right|\right) \tag{3}$$

$$\leq P\left(|S_{dn} - E_{dn}| > \frac{dn}{2} - \frac{dn}{4}\right) \tag{4}$$

$$\leq P\left(|S_{dn} - E_{dn}| > \frac{dn}{4}\right) = 2e^{-\frac{dn}{8}} \tag{5}$$

(3) to (4) is due the fact that for all valid values of $\mathcal{U}$, it holds that $E_{dn} = dn(\frac{\mathcal{U}}{n}) \geq \frac{dn}{2}$. (5) is due to Hoeffding's inequality. As such, $P(\varepsilon) \leq 2e^{-\frac{dn}{8}}$. Thus, with high probability there are at least $\frac{dn}{4}$ balls that fall into bins corresponding to $\mathcal{U}$. Since balls are thrown independently and uniformly at random, conditioned on the balls that land in $\mathcal{U}$, they are thrown independently and uniformly at random. $\qquad\square$

## D  Proof of Lemma 3

Recall that $\mu = \sum_{i \in S'} p(X = i)$. We have

$$H(X) \geq \sum_{i \in S'} p(X = i) \log\left(\frac{1}{p(X = i)}\right)$$

$$= \mu\left(\sum_{i \in S'} \frac{p(X = i)}{\mu} \log\left(\frac{1}{p(X = i)}\right)\right)$$

$$\geq \mu\left(\sum_{i \in S'} \frac{p(X = i)}{\mu} \log\left(\frac{\mu}{p(X = i)}\right)\right)$$

$$= \mu\left(\sum_{i \in S'} p(X' = i) \log\left(\frac{1}{p(X' = i)}\right)\right)$$

$$= \mu H(X'). \qquad\square$$

## E  Proof of Lemma 5

We show the contrapositive. Suppose that $p_1 \leq \varepsilon < 2^{-c}$. We have $p_i \leq p_1, \forall i \in [m]$. We consider all such distributions and find the one with smallest entropy:

$$\min_{p_1 \geq p_2, \ldots \geq p_m} H([p_1, p_2, \ldots, p_m])$$

$$\text{s.t.} \quad \sum_i p_i = 1 \tag{6}$$

$$\varepsilon \geq p_i \geq 0, \forall i \in [m]$$

For simplicity, suppose $\frac{1}{\varepsilon}$ is an integer. We show that the solution to the above optimization problem is strictly greater than $c$ using majorization theory. For any given $p$, define the vector $u_p = [\sum_{j=1}^{i} p_j]_i$. Recall that a probability distribution $p$ majorizes another distribution $q$ if $u_p(i) \geq u_q(i), \forall i \in [m]$. Also if $p$ majorizes $q$, we have $H(p) \leq H(q)$.

Consider all distributions in the feasible region of the above problem. For any $p^*$, consider the vector $u_{p^*}$. Clearly, $u_{p^*}(1) \geq \varepsilon$. Since $p_2 \leq p_1 < \varepsilon$, we have that $u_{p^*}(2) \leq 2\varepsilon$. Similarly, we have $u_{p^*}(i) \leq \varepsilon$. The uniform distribution achieves this upper bounding $u$ vector, establishing that the uniform distribution majorizes every other distribution in the feasible set. Then for any distribution in the feasible region, we get that $H(p) \geq \log(\frac{1}{\varepsilon}) > c$.

Suppose $\frac{1}{\varepsilon}$ is not an integer. Let $t$ be the largest integer such that $t\varepsilon \leq 1$. Then the above argument leads to the distribution with entropy

$$H = t\varepsilon \log\left(\frac{1}{\varepsilon}\right) + (1 - t\varepsilon)\log\left(\frac{1}{1 - t\varepsilon}\right). \tag{7}$$

Next, we show that if $\varepsilon < 2^{-c}$, above value is greater than $c$. We can rewrite

$$H = t\varepsilon \log\left(\frac{1}{\varepsilon}\right) + (1 - t\varepsilon)\log\left(\frac{1}{1 - t\varepsilon}\right) \tag{8}$$

$$\geq t\varepsilon \log\left(\frac{1}{\varepsilon}\right) + (1 - t\varepsilon)\log\left(\frac{1}{\varepsilon}\right) \tag{9}$$

$$= \log\left(\frac{1}{\varepsilon}\right) > c \tag{10}$$

since $1 - t\varepsilon \leq \varepsilon$. This concludes the proof. $\square$

## F   Proof of Proposition 1

By Lemma 5 we then know $\rho' \leq \frac{\rho}{e_1} \leq \rho 2^c$, and the size of the support of $X_{S',2}$ is the number of balls in the most loaded bin which is at least $\frac{\ln(n) - \ln(2)}{\ln(\ln(n)) + \ln(\frac{4}{d})}$. Using Lemma 4, we conclude $H(X_{S',2}) \geq \log\left(\frac{\ln(n) - \ln(2)}{\ln(\ln(n)) + \ln(\frac{4}{d})}\right) - \left(\frac{\rho 2^c \ln(\rho 2^c)}{\rho 2^c - 1} - 1 - \ln\left(\frac{\rho 2^c \ln(\rho 2^c)}{\rho 2^c - 1}\right)\right)\frac{1}{\ln(2)}$.

Using our previous results, we know that $\min_{i \in S'} p(X = i, Y = 2) \geq \min_{i \in S'} e_1 x_i \geq \frac{e_1}{\sqrt{\rho n}} \geq \frac{2^{-c}}{\sqrt{\rho n}}$.
Then, $p(X \in S', Y = 2) \geq \left(\frac{\ln(n) - \ln(2)}{\ln(\ln(n)) + \ln(\frac{4}{d})}\right)\left(\frac{2^{-c}}{\sqrt{\rho n}}\right) = \frac{\ln(n) - \ln(2)}{(\ln(\ln(n)) + \ln(\frac{4}{d}))\sqrt{\rho n}2^c}$. Additionally:

$$p(X \notin S', Y = 2) = \sum_{i \in S^c}\sum_{j=1}^{m} p(X = i, Y = 2, E = j)$$

$$+ \sum_{i \in S, i \notin S'}\sum_{j=1}^{m} p(X = i, Y = 2, E = j) \tag{11}$$

$$= \sum_{i \in S^c}\sum_{j=1}^{m} p(X = i, Y = 2, E = j)$$

$$+ \sum_{i \in S, i \notin S'}\sum_{j=2}^{m} p(X = i, Y = 2, E = j) \tag{12}$$

$$\leq \sum_{i \in S^c}\sum_{j=1}^{m} p(X = i, Y = 2, E = j)$$

$$+ \sum_{i \in S}\sum_{j=2}^{m} p(X = i, Y = 2, E = j) \leq \frac{2}{n}. \tag{13}$$

We go from (11) to (12) by realizing that for any $i \in S$, $p(X = i, Y = 2, E = 1) > 0$ only if $M_{x,1} = 2$ and thus $i \in S'$. We simplify (12) by definition of $\mathcal{U}$. As such, $p(X \in S'|Y = 2) = \frac{p(X \in S', Y = 2)}{p(X \in S', Y = 2) + p(X \notin S', Y = 2)} \geq \frac{\frac{\ln(n) - \ln(2)}{(\ln(\ln(n)) + \ln(\frac{4}{d}))\sqrt{\rho n}2^c}}{\frac{\ln(n) - \ln(2)}{(\ln(\ln(n)) + \ln(\frac{4}{d}))\sqrt{\rho n}2^c} + \frac{2}{n}} = \frac{\ln(n) - \ln(2)}{\ln(n) - \ln(2) + (\ln(\ln(n)) + \ln(\frac{4}{d}))\sqrt{\rho}2^{c+1}}$.
Thus, we have shown that $H(X_{S',2}) \geq \log\left(\frac{\ln(n) - \ln(2)}{\ln(\ln(n)) + \ln(\frac{4}{d})}\right) - \left(\frac{\rho 2^c \ln(\rho 2^c)}{\rho 2^c - 1} - 1 - \ln\left(\frac{\rho 2^c \ln(\rho 2^c)}{\rho 2^c - 1}\right)\right)\frac{1}{\ln(2)}$
and $P(X \in S', Y = 2) \geq \frac{\ln(n) - \ln(2)}{\ln(n) - \ln(2) + (\ln(\ln(n)) + \ln(\frac{4}{d}))\sqrt{\rho}2^{c+1}}$.

Using Lemma 3 we have:

$$H(\tilde{E}) \geq H(X|Y=2) \geq P(X \in S', Y=2)(H(X_{S',2}))$$

$$\geq \left( \frac{\ln(n) - \ln(2)}{\ln(n) - \ln(2) + (\ln(\ln(n)) + \ln(\frac{4}{d}))\sqrt{\rho}2^{c+1}} \right)$$

$$\left( \log\left( \frac{\ln(n) - \ln(2)}{\ln(\ln(n)) + \ln(\frac{4}{d})} \right) \right.$$

$$\left. - \left( \frac{\rho 2^c \ln(\rho 2^c)}{\rho 2^c - 1} - 1 - \ln\left( \frac{\rho 2^c \ln(\rho 2^c)}{\rho 2^c - 1} \right) \right) \frac{1}{\ln(2)} \right)$$

$$= \left( 1 - \frac{(\ln(\ln(n)) + \ln(\frac{4}{d}))\sqrt{\rho}2^{c+1}}{\ln(n) - \ln(2) + (\ln(\ln(n)) + \ln(\frac{4}{d}))\sqrt{\rho}2^{c+1}} \right)$$

$$\left( \log\left( \frac{\ln(n) - \ln(2)}{\ln(\ln(n)) + \ln(\frac{4}{d})} \right) \right.$$

$$\left. - \left( \frac{\rho 2^c \ln(\rho 2^c)}{\rho 2^c - 1} - 1 - \ln\left( \frac{\rho 2^c \ln(\rho 2^c)}{\rho 2^c - 1} \right) \right) \frac{1}{\ln(2)} \right). \tag{14}$$

Since $c = O(1)$ and $d = \Theta(1)$, this lower bound is asymptotically $H(\tilde{E}) \geq \max_y H(X|Y=y) \geq H(X|Y=2) \geq (1 - o(1))(\log(\log(n)) - \log(\log(\log(n)))) - \mathcal{O}(1))$.

Now when $n \geq \nu(r, q, \rho, c, d)$, we can lower bound the $(1 - o(1))$ term as:

$$1 - \frac{(\ln(\ln(n)) + \ln(\frac{4}{d}))\sqrt{\rho}2^{c+1}}{\ln(n) - \ln(2) + (\ln(\ln(n)) + \ln(\frac{4}{d}))\sqrt{\rho}2^{c+1}} \tag{15}$$

$$\geq 1 - \frac{(1+r)\ln(\ln(n))\sqrt{\rho}2^{c+1}}{\ln(n/2) + \ln(\ln(n))\sqrt{\rho}2^{c+1}} \tag{16}$$

$$\geq 1 - \frac{(1+r)\sqrt{\ln(n/2)}\sqrt{\rho}2^{c+1}}{\ln(n/2) + \sqrt{\ln(n/2)}\sqrt{\rho}2^{c+1}} \tag{17}$$

$$= 1 - \frac{1+r}{1 + \frac{\ln(n/2)}{\sqrt{\rho}2^{c+1}}} \tag{18}$$

$$\geq 1 - \frac{1+r}{1+q} \tag{19}$$

We bound from (15) to (16) by using $n \geq e^{\left(\frac{4}{d}\right)^{1/r}}$ which implies $\ln(\ln(n)) + \ln(\frac{4}{d}) \leq (1+r)\ln(\ln(n))$. We go from (16) to (17) by using $\sqrt{\ln(n/2)} \geq \ln(\ln(n))$ when $n \geq 3$. We bound from (18) to (19) by using $n \geq 2e^{q^2 2^{2(c+1)}\rho}$. Next, we lower bound the term $\log\left(\frac{\ln(n) - \ln(2)}{\ln(\ln(n)) + \ln(\frac{4}{d})}\right)$.

$$\log\left( \frac{\ln(n) - \ln(2)}{\ln(\ln(n)) + \ln(\frac{4}{d})} \right) \tag{20}$$

$$\geq \log\left( \frac{\ln(n/2)}{(1+r)\ln(\ln(n))} \right) \tag{21}$$

$$\geq \log\left( \sqrt{\ln(n/2)} \right) - \log(1+r) \tag{22}$$

$$\geq 0.5\log(0.5\log(n/2)) - \log(1+r) \tag{23}$$

$$\geq 0.5\log(\log(n)) - \log(1+r) - 1 \tag{24}$$

We bound from (20) to (21) by using $\ln(\ln(n)) + \ln(\frac{4}{d}) \leq (1+r)\ln(\ln(n))$. We bound from (21) to (22) using $\sqrt{\ln(n/2)} \geq \ln(\ln(n))$. We then substitute all of these bounds into our previous lower

bound on $H(\tilde{E})$ (14) yielding:

$$H(\tilde{E}) \geq \left(1 - \frac{1+r}{1+q}\right)\left(0.5\log\left(\log\left(n\right)\right) - \log\left(1+r\right)\right.$$

$$-\mathcal{O}(1) - \frac{1}{\ln(2)}\left(\frac{\rho 2^c \ln(\rho 2^c)}{\rho 2^c - 1} - 1 - \ln\left(\frac{\rho 2^c \ln\left(\rho 2^c\right)}{\rho 2^c - 1}\right)\right)\right)$$

$$= \left(1 - \frac{1+r}{1+q}\right)\left(0.5\log(\log(n)) - \log(1+r) - \mathcal{O}(1)\right).$$

## G    Proof of Corollary 1

### G.0.1    Condition (a): Bounded Ratio

We know that $\frac{\max_x p(x)}{\min_x p(x)} \leq \rho$. Since $\sum_x p(x) = 1$, $\min_x p(x) \leq \frac{1}{n} \leq \max_x p(x)$ and we have $\frac{\max_x p(x)}{1/n} \leq \rho \Rightarrow \max_x p(x) \leq \frac{\rho}{n}$ and similarly $\min_x p(x) \geq \frac{1}{\rho n}$. Then using Theorem 1, when $n \geq \nu(r=1, q=3, \rho^2, c, d=1)$, $H(\tilde{E}) \geq \max_y H(X|Y=y) \geq 0.25\log\left(\log\left(n\right)\right) - \mathcal{O}(1)$ with high probability (where the $\mathcal{O}(1)$ term is a function of only $\rho, c$). As such, there exists an $n_0$ (which is a function of only $\rho, c$) such that for all $n > n_0$, the causal direction is identifiable with high probability.

### G.1    Condition (b): Sampled Uniformly on the Simplex

We first show that when the distribution of $X$ is uniformly sampled from the simplex, there exist a set $S$ that satisfies the assumptions of Theorem 1 with high probability.

**Lemma 6.** *When the $x_i$ are sampled uniformly from the simplex, there exists a subset of the support with size at least $(e^{-\frac{1}{\sqrt{\rho}}} - \frac{1}{\sqrt{\rho}} - \delta)n$ for which all $x_i$ are within a factor of $\sqrt{\rho}$ from $\frac{1}{n}$ and make up total probability mass $\geq \left(e^{-\frac{1}{\sqrt{\rho}}} - \frac{1}{\sqrt{\rho}} - \delta\right)\frac{1}{\sqrt{\rho}}$, with probability $> 1 - 2e^{-2\delta^2 n}$ for $\rho, n \geq 1$, $\delta > 0$.*

*Proof.* Let us call a probability "small" if $x_i \leq \frac{1}{\sqrt{\rho}n}$. We want to show that with high probability (at least $1 - 2e^{-2\delta^2 n}$), there are at most $(1 - e^{-\frac{1}{\sqrt{\rho}}} + \delta)n$ small $x_i$. Using Theorem 3 of [19], we know that for each $x_i$ in a Dirichlet distribution with $\alpha = 1$ (i.e., the uniform distribution over the probability simplex) and support size $n$, $P(x_i > z) = (1 - z)^{n-1}$ (This is by setting $a_i = z$ and $a_j = 0, \forall j \neq i$ and using the fact that $P(x_i = 0) = 0, \forall i \in [n]$). As such, $P(x_i \leq z) = 1 - (1 - z)^{n-1}$. The probability that $x_i$ is small is then equal to $P(x_i \leq \frac{1}{\sqrt{\rho}n}) = 1 - (1 - \frac{1}{\sqrt{\rho}n})^{n-1}$. This value is non-decreasing when $n \geq 1$, and approaches $1 - e^{-\frac{1}{\sqrt{\rho}}}$ as $n$ approaches infinity. Hence when $n \geq 1$, the probability that any $x_i$ is "small" is upper-bounded by $1 - e^{-\frac{1}{\sqrt{\rho}}}$. We want to show that the outcome that there are more than $(1 - e^{-\frac{1}{\sqrt{\rho}}} + \delta)n$ small $x_i$ will not happen with high probability. To do this, we note that all $x_i$ in a symmetric Dirichlet distribution are negatively associated (this follows from Lemma 9 in Section N). This implies that the probability that there are at least $(1 - e^{-\frac{1}{\sqrt{\rho}}} + \delta)n$ small $x_i$ is upper-bounded by the probability that there are at least that many $x_i$ when we treat the $x_i$ as if they are i.i.d. random variables. This allows us to use Hoeffding's inequality. Let $S_n$ be the total number of small $x_i$ and $E_n$ be the expected number of small $x_i$. Since $E_n \leq (1 - e^{-\frac{1}{\sqrt{\rho}}})n$, then $P(S_n > (1 - e^{-\frac{1}{\sqrt{\rho}}} + \delta)n) \leq P(|S_n - E_n| > \delta n) < 2e^{-2\delta^2 n}$. As such, the probability that there are at most $(1 - e^{-\frac{1}{\sqrt{\rho}}} + \delta)n$ small $x_i$ is at least $(1 - 2e^{-2\delta^2 n})$.

Let us call an $x_i$ "big" if $x_i \geq \frac{\sqrt{\rho}}{n}$. There are at most $\frac{n}{\sqrt{\rho}}$ big $x_i$, since otherwise their total probability mass would exceed 1.

Next, consider the subset of $x_i$ that are neither "big" nor "small". They are in the range $[\frac{1}{\sqrt{\rho}n}, \frac{\sqrt{\rho}}{n}]$. We know that with high probability $(1 - 2e^{-2\delta^2 n})$ there are at most $(1 - e^{-\frac{1}{\sqrt{\rho}}} + \delta)n$ small $x_i$ and

at most $\frac{n}{\sqrt{\rho}}$ big $x_i$. This means our desired subset has size at least $\left(n - (1 - e^{-\frac{1}{\sqrt{\rho}}} + \delta)n - \frac{n}{\sqrt{\rho}}\right) = \left(e^{-\frac{1}{\sqrt{\rho}}} - \frac{1}{\sqrt{\rho}} - \delta\right)n$ with probability at least $1 - 2e^{-2\delta^2 n}$. $\qquad\square$

As such, if we set $\rho = 25$ and $\delta = 0.1$, there exists a subset of the support of size $\geq (e^{-\frac{1}{\sqrt{25}}} - \frac{1}{\sqrt{25}} - 0.1)n \geq 0.5n$ where all $x_i$ are within a factor of $\sqrt{25} = 5$ from $\frac{1}{n}$ with probability $> 1 - 2e^{-2(0.1)^2 n} = 1 - 2e^{-0.02n}$. Using Theorem 1, we conclude that when $n \geq \nu(r = 1, q = 3, \rho = 25, c, d = 0.5)$, $H(\tilde{E}) \geq \max_y H(X|Y = y) \geq 0.25 \log(\log(n)) - \mathcal{O}(1)$ with high probability (where the $\mathcal{O}(1)$ term is a function of only $c$). As such, there exists an $n_0$ (which is a function of only $c$) such that for all $n > n_0$, the causal direction is identifiable with high probability.

### G.2 Condition (c): High Entropy

We show that when $X$ has entropy within an additive constant of $\log(n)$, there exists a set $S$ that satisfies the assumptions of Theorem 1.

**Lemma 7.** *For any distribution $X$ with support size $n$ and entropy $\geq \log(n) - a$, there exists a subset $S$ with all $x_i \in .[\frac{3}{40n}, \frac{2^{2b}}{n}]$ for $i \in S$, and support size $|S| \geq \frac{n}{2^{2b+3}}$, where $b = \max\{a, 2\}$.*

*Proof.* Let us call an $x_i$ "large" if $x_i \geq \frac{2^{2b}}{n}$, and $\mu_{\text{large}}$ be the total probability mass contributed by large $x_i$. The upper bound for the sum of the terms in the formula for $H(X)$ corresponding to large $x_i$ is $\mu_{\text{large}} \log(\frac{n}{2^{2b}})$. The upper bound for the sum of the terms in Shannon entropy corresponding to $x_i$ that are not large is $(1 - \mu_{\text{large}}) \log(\frac{n}{1-\mu_{\text{large}}})$. Since entropy is greater than $\log(n) - a$ and $b = \max\{a, 2\}$, we have that entropy is greater than or equal to $\log(n) - b$ as well. Then, for the total entropy to be at least $\log(n) - b$ it must be true that $\mu_{\text{large}} \log(\frac{n}{2^{2b}}) + (1 - \mu_{\text{large}}) \log(\frac{n}{1-\mu_{\text{large}}}) \geq \log(n) - b$. It follows that $2b\mu_{\text{large}} + (1 - \mu_{\text{large}}) \log(1 - \mu_{\text{large}}) \leq b$. For $x \geq 0$, we have that $(1-x)\log(1-x) \geq -1.5x$. Then we have $2\mu_{\text{large}}(b - 0.75) \leq b$, or equivalently $\mu_{\text{large}} \leq \frac{b}{2(b-0.75)}$. Since $b \geq 2$, we have that $\mu_{\text{large}} \leq 0.8$.

Let us call an $x_i$ "small" if it is $\leq \frac{0.075}{n}$, and let $\mu_{\text{small}}$ be the total probability mass in small $x_i$. Even if all $x_i$ were small (although that would be impossible), $\mu_{\text{small}} \leq 0.075$. As such, $\mu_{\text{small}} + \mu_{\text{large}} \leq \frac{7}{8}$. This means at least $\frac{1}{8}$ total probability mass belongs to $x_i \in [\frac{0.075}{n}, \frac{2^{2b}}{n}]$. Our subset $S$ of $X$ will be all of these $x_i$. Since every element in $X$ is upper-bounded by $\frac{2^{2b}}{n}$, $S$ has a support size of at least $\frac{\frac{1}{8}}{\frac{2^{2b}}{n}} = \frac{n}{2^{2b+3}}$. $\qquad\square$

We can therefore satisfy the conditions of Theorem 1 with $d = \frac{1}{2^{2\max\{a,2\}+3}}$ and $\rho \leq (\frac{40}{3}2^{2\max\{a,2\}})^2 \leq 2^{4\max\{a,2\}+8}$. Using Theorem 1, we conclude that when $n \geq \nu(r = 1, q = 3, \rho = 2^{4\max\{a,2\}+8}, c, d = \frac{1}{2^{2\max\{a,2\}+3}})$, $H(\tilde{E}) \geq \max_y H(X|Y = y) \geq 0.25 \log(\log(n)) - \mathcal{O}(1)$ with high probability (where the $\mathcal{O}(1)$ term is a function of only $a, c$). Hence there exists an $n_0$ (a function of only $a, c$) such that for all $n > n_0$, the causal direction is identifiable with high probability.

## H Proof of Theorem 2

Given the random variables $U_i, i \in [n]$ with marginal distributions $\mathbf{p_i}(u_i)$, let $p(u_1, u_2, \ldots, u_n)$ be a valid coupling. Then $p$ satisfies $\mathbf{p_i}(u_i) = \sum_{k \neq i} \sum_{u_k \in [n]} p(u_1, u_2, \ldots, u_n)$ holds for all $i, u_i$. Therefore, for all $i, u_i$, we can define $S_{i,u_i} = \{(u_j)_{j \neq i} : p(u_1, u_2, \ldots u_n) > 0\}$. $S_{i,u_i}$ contains the coordinates in the coupling that contribute non-zero mass to satisfy the $i^{th}$ marginal distribution, specifically the probability that variable $U_i$ takes the value $u_i$. Let us define the function $g_{i,u_i}((u_j)_{j \neq i}) := p(u_1, \ldots, u_n)$. Then equivalently, we can write $\mathbf{p_i}(u_i) = \sum_{t \in S_{i,u_i}} g_{i,u_i}(t)$.

Consider a noisy version of the marginal distributions: Let $\hat{\mathbf{p}}_{\mathbf{i}}$ be the noisy marginals where $|\hat{\mathbf{p}}_{\mathbf{i}}(u_i) - \mathbf{p_i}(u_i)| \leq \delta$ for all $i, u_i$. Our strategy is to start with the coupling $p(u_1, \ldots, u_n)$ and convert it to a

---

**Algorithm 1** Phase I

---

**Input:** Valid coupling $p_{\text{init}}$ for the marginals $\{\mathbf{p_i}\}_{i \in [n]}$. Noisy marginals $\{\hat{\mathbf{p}}_{\mathbf{i}}\}$

$p \leftarrow p_{\text{init}}$.

Construct $g_{i,u_i}, S_{i,u_i}, T_i^+, T_i^-$ from $p_{\text{init}}$ for all $i, u_i$.

**while** $\exists i \in [n]$ s.t. $T_i^- \neq \emptyset$ **do**
   Pick arbitrary $h_{i,u_i}$ for all $u_i$ such that

$$0 \le h_{i,u_i}(t) \le g_{i,u_i}(t), \forall t \in S_{i,u_i},$$

$$\sum_{t \in S_{i,u_i}} h_{i,u_i}(t) = \hat{\mathbf{p}}_{\mathbf{i}}(u_i).$$

   Update $p$ as follows:

$$p(u_1, u_2, \ldots, u_n) \leftarrow h_{i,u_i}((u_j)_{j \neq i}), \forall (u_j)_{j \neq i} \in S_{i,u_i} \tag{27}$$

   Construct $g_{i,u_i}, S_{i,u_i}, T_i^+, T_i^-$ from $p$ for all $i, u_i$.
**end while**
return $p$

---

coupling for the noisy marginals. Let us define $T_i^+(p) := \{u_i : \sum_{k \neq i} \sum_{u_k \in [n]} p(u_1, u_2, \ldots, u_n) < \hat{\mathbf{p}}_{\mathbf{i}}(u_i)\}, T_i^-(p) := \{u_i : \sum_{k \neq i} \sum_{u_k \in [n]} p(u_1, u_2, \ldots, u_n) > \hat{\mathbf{p}}_{\mathbf{i}}(u_i)\}$. In words, $T_i^+(p)$ shows the coordinates of the $i^{th}$ noisy marginal which has excess mass compared to the mass induced by coupling $p$. Similarly, $T_i^-(p)$ shows the coordinates of the $i^{th}$ noisy marginal for which the coupling $p$ has more mass than needed. We update $p$ in two stages: First, we update $p$ so that $T_i^-(p) = \emptyset$. In the second stage, we further update $p$ so that $T_i^+(p) = \emptyset$ and $T_i^-(p) = \emptyset$, which shows that the updated $p$ is a valid coupling for the noisy marginals $\hat{\mathbf{p}}_{\mathbf{i}}$. We finally bound the entropy of the new coupling relative to the initial coupling we started with.

First we observe the following: Consider any $u_i \in T_i^-$. Then there exists a function $h_{i,u_i}(t)$ such that

$$0 \le h_{i,u_i}(t) \le g_{i,u_i}(t), \forall t \in S_{i,u_i}, \tag{25}$$

$$\sum_{t \in S_{i,u_i}} h_{i,u_i}(t) = \hat{\mathbf{p}}_{\mathbf{i}}(u_i). \tag{26}$$

This is true since $\sum_{t \in S_{i,u_i}} g_{i,u_i}(t) = \mathbf{p_i}(u_i)$ and $\hat{\mathbf{p}}_{\mathbf{i}}(u_i) < \mathbf{p_i}(u_i), \forall u_i \in T_i$. We can describe the first phase as follows: For each $i \in [n]$ and $u_i \in T_i^-$, we pick an arbitrary $h_{i,u_i}$ and update $p$ to match the entries of $h_{i,u_i}$. Notice that each update of $p$ changes the corresponding $h, g$ functions. Our construction proceeds by updating these functions every time $p$ is updated as given above. This procedure is summarized in Algorithm 1.

Note that the size of $T_i^-$ after an update is at least one less than the size of $T_i^-$ before the update. To see this, note that after the update in (27), $u_i \notin T_i^-$. Also by reducing elements of $p$, we can never add a new element to $T_i^-$ for any $i$ by definition of $T_i^-$. Therefore, after at most $\sum_i |T_i^-|$ applications of the above update for the initial sets $T_i^-$, we have $T_i^- = \emptyset, \forall i \in [n]$. Since there are at most $n$ elements in $T_i^-$ and $n$ such sets, the first phase terminates in at most $n^2$ steps.

Let $p$ be the output of Algorithm 1 in the rest of the proof. In the second phase, we consider the updated $T_i^+$. Our strategy here is to distribute the remaining mass in each marginal as its own coupling and add this coupling to $p$ that is the output of Algorithm 1. Let us represent the excess probability mass in coordinate $u_i$ of marginal $i$ relative to coupling $p$ by $r_{i,u_i}$. Note that $r_{i,u_i}(p) := \hat{\mathbf{p}}_{\mathbf{i}}(u_i) - \sum_{k \neq i} \sum_{u_k \in [n]} p(u_1, u_2, \ldots, u_n)$ may increase at each step of the first phase. The exact increase in this gap for each $i, u_i$ depends on the choice of $h_{i,u_i}$ function at each step. However, we can bound the total gap per marginal at the end of first phase as $\sum_{u_i \in [n]} r_{i,u_i}(p) \le \delta n^2, \forall i$. Each step of Algorithm 1 can add a mass of at most $\delta$ to each marginal at each step (it terminates after at most $\sum_i |T_i^-|$ steps) and at the beginning of first phase, each coordinate of each marginal has at most $\delta$ excess mass (there are $\sum_i |T_i^+|$ coordinates with excess mass). As such, there is at most

---

$\sum_i \delta |T_i^-| + \sum_i \delta |T_i^+| \leq \delta n^2$ total gap per marginal at the end of the first phase. Let $p(u_1, \ldots, u_n)$ be the output of Algorithm 1. [15] showed a greedy minimum entropy coupling algorithm that produces a coupling with support at most $n^2$. Let $q(u_1, u_2 \ldots, u_n)$ be the output of this greedy algorithm when given the excess marginal mass as its input. Then we have that $v := p + q$ is a valid coupling for the noisy marginals. This is because, by feeding the greedy algorithm the excess marginal mass, we guarantee that the marginals of $v$ are correct. Moreover, all cells in the coupling are in range $[0, 1]$ as no cell in $p$ or $q$ has negative value and their sum has the correct marginals.

Next, define the distribution $s : 2 \times [n]^n \to [0, 1]$ as follows:

$$s(0, u_1, u_2, \ldots, u_n) = p(u_1, u_2, \ldots, u_n), \tag{28}$$
$$s(1, u_1, u_2, \ldots, u_n) = q(u_1, u_2 \ldots, u_n). \tag{29}$$

From the argument above, it is easy to see that $s$ is a valid probability distribution, i.e., it has non-negative entries and its entries sum to 1.

We compare entropy of the obtained coupling $v$ with entropy of $s$ and that with entropy of the initial coupling $p_{\text{init}}$. First, it is easy to see from concavity of entropy and Jensen's inequality that $H(v) \leq H(s)$. Let $\bar{H}$ be the extended entropy operator that admits vectors outside the simplex as input, for vectors whose entries are between 0 and 1: $\bar{H}(p(x)) = -\sum_x p(x) \log(p(x))$. We have the following lemma that allows us to compare $\bar{H}(p)$ with $H(p_{\text{init}})$:

**Lemma 8.** *Let $\mathbf{p} = [p_1, p_2, \ldots, p_n]$ be a discrete probability distribution. Let $\mathbf{q} = [q_1, q_2, \ldots, q_n]$ be a non-negative vector such that $q_i \leq p_i, \forall i \in [n]$. Then $\bar{H}(\mathbf{q}) \leq \bar{H}(\mathbf{p}) + \frac{\log(e)}{e}$.*

The proof is in Section I in the supplement.

From the lemma, we can conclude that $\bar{H}(p) \leq H(p_{\text{init}}) + \frac{\log(e)}{e}$. Finally, the maximum entropy contribution of $q$ is when it induces uniform distribution over $n^2$ states. Since the total mass of $q$ is $\delta n^2$, we have

$$\bar{H}(q) \leq n^2 \left( \frac{\delta n^2}{n^2} \log \left( \frac{n^2}{\delta n^2} \right) \right) \tag{30}$$

$$= \delta n^2 \log \left( \frac{1}{\delta} \right) \tag{31}$$

Suppose $\delta \leq \frac{1}{n^2 \log(n)}$. Then we can further bound $\bar{H}(q) \leq 2 + \frac{\log(\log(n))}{\log(n)} \leq 2 + o(1)$ since $\delta \log \left( \frac{1}{\delta} \right) \leq \frac{2 \log(n) + \log(\log(n))}{n^2 \log(n)}$ if $\delta < \frac{1}{n^2 \log(n)}$.

Bringing it all together, we obtain the following chain of inequalities:

$$H(v) \leq H(s) = \bar{H}(p) + \bar{H}(q) \tag{32}$$

$$\leq H(p_{\text{init}}) + \frac{\log(e)}{e} + 2 + o(1). \tag{33}$$

This concludes the proof. $\qquad \square$

## I   Proof of Lemma 8

If $p_i < \frac{1}{\exp(1)}, \forall i$, due to monotonicity of $-p \log(p)$ in $p$, we have $\bar{H}(\mathbf{q}) \leq \bar{H}(\mathbf{p})$.

In general, no more than 2 states can satisfy $p_i > \frac{1}{\exp(1)}$. Therefore, $\bar{H}(q)$ can only be larger than $\bar{H}(p)$ due to two states. Let us call these two states $p_1, p_2$ without loss of generality. Reducing the probability of any other state only gives a looser bound.

We can obtain the largest entropy increase by solving the following optimization problem:

$$\max_{p_1, p_2} \quad \mathbb{1}_{\{p_1 > 1/e\}} \left( \frac{\log(e)}{e} - p_1 \log \left( \frac{1}{p_1} \right) \right)$$
$$+ \mathbb{1}_{\{p_2 > 1/e\}} \left( \frac{\log(e)}{e} - p_2 \log \left( \frac{1}{p_2} \right) \right) \tag{34}$$
$$\text{subject to} \quad p_1 + p_2 \leq 1,$$
$$p_1 \geq 0, p_2 \geq 0$$

Suppose $p_1 > 1/e$ and $p_2 < 1/e$. Then the solution is simply to set $p_1 = 1$ since this minimizes the entropy contribution of $p_1$. This gives a gap of $\frac{\log(e)}{e}$. Due to symmetry, we only need to investigate the case where $p_1 > 1/e$ and $p_2 > 1/e$. In this case, we have the following optimization problem:

$$
\begin{aligned}
\min_{p_1, p_2} \quad & p_1 \log\left(\frac{1}{p_1}\right) + p_2 \log\left(\frac{1}{p_2}\right) \\
\text{subject to} \quad & p_1 + p_2 \leq 1, \\
& p_1 \geq 1/e, p_2 \geq 1/e
\end{aligned}
\tag{35}
$$

This is a concave minimization problem and the solution has to be at the boundary of the convex constraint region. If $p_1 = 1/e$, the maximum gap is obtained when $p_2$ is maximized to $p_2 = 1 - 1/e$ which gives a gap that is strictly less than $\frac{\log(e)}{e}$, hence we can discard this solution for the maximum entropy gap. $p_2 = 1/e$ gives the same solution from symmetry. When $p_1 + p_2 = 1$, the problem reduces to minimizing the binary entropy function, which again is minimized at the boundary. The boundary in this case is where either $p_1 = 1/e$ or $p_2 = 1/e$. Therefore, both probabilities being greater than $1/e$ cannot yield a better bound. □

## J   Proof of Lemma 1

**Joint Probabilities**. First, we bound the estimates of the entries of the joint distribution between $X$ and $Y$. Both $X$ and $Y$ have $n$ states which we index as $i = 1, \ldots, n$ and $j = 1, \ldots, n$ respectively. Hence the joint distribution has $n^2$ states. Probability that $X = i$ and $Y = j$ is shown as $p_{ij}$. Suppose $N$ samples from $N$ independent, identically distributed random variables are drawn as $\{(x_k, y_k)\}_{k \in [N]}$. This yields the empirical probability estimates ($I$ is the indicator function)

$$
\hat{p}_{ij} = \frac{1}{N} \sum_{k=1}^{N} I(x_k = i \,\&\, y_k = j).
$$

Note that each of these estimates are averages of Bernoulli random variables with success probability $p_{ij}$. We also consider the marginal probability empirical estimates

$$
\hat{p}_i^X = \frac{1}{N} \sum_{k=1}^{N} I(x_k = i)
$$

and

$$
\hat{p}_j^Y = \frac{1}{N} \sum_{k=1}^{N} I(y_k = j).
$$

which are also averages of $N$ Bernoulli random variables (with success probabilities $p_i^X$ and $p_j^Y$ respectively).

Since these estimates are clearly correlated with one another, our approach will be to use concentration results on individual entries of the joint distribution and then do a union bound over all $n^2 + 2n$ probabilities. Note that $I(x_k = i \,\&\, y_k = j) = 1$ with probability $p_{ij}$ and 0 otherwise. Thus by Hoeffding's inequality [31],

$$
\mathbb{P}\left\{|\hat{p}_{ij} - p_{ij}| \geq t\right\} \leq 2 \exp\left(-2t^2 N\right).
\tag{36}
$$

We can define an event $\mathcal{A}$ where all the probability estimates are within $t$ of the truth:

$$
\mathcal{A} = \left\{ \max_{i,j \in 1,\ldots,n} |\hat{p}_{ij} - p_{ij}| \leq t \right\} \bigcap \left\{ \max_{i \in 1,\ldots,n} |\hat{p}_i^X - p_i^X| \leq t \right\} \bigcap \left\{ \max_{j \in 1,\ldots,n} |\hat{p}_j^Y - p_j^Y| \leq t \right\}.
$$

Starting with (36) and taking the union bound over all $n^2 + 2n$ probabilities in the joint and marginal distribution, we obtain

$$
\begin{aligned}
\mathbb{P}(\mathcal{A}) &> 1 - 2(n^2 + 2n) \exp\left(-2t^2 N\right) \\
&> 1 - 4 \exp(2 \ln(n) - 2t^2 N).
\end{aligned}
\tag{37}
$$

**Conditional Probabilities**. Given the above bound on the estimates of the joint probabilities, we formulate bounds on the conditional probability estimates. Recall that

$$P(X = i | Y = j) = \frac{P(X = i, Y = j)}{P(Y = j)} = \frac{p_{ij}}{\sum_{i=1}^{n} p_{ij}}.$$

Using the plug-in approach, we have

$$\hat{p}_{i|j} = \frac{\hat{p}_{ij}}{\hat{p}_j^Y}.$$

Note that it is critical for $\hat{p}_j^Y$ to be bounded away from zero, otherwise a small error in $\hat{p}_{ij}$ may cause a large error in $\hat{p}_{i|j}$. In what follows, we set

$$\alpha = \frac{\min_{j=1,\dots,n} p_j^Y}{2}.$$

$\alpha$ will naturally appear in the number of samples, and notably must depend on $n$. Note that the case of $\sum_{i=1}^{n} p_{ij} = 0$ is allowable since if that is the case $Y = j$ will never occur and corresponding probability estimates will all be zero and the conditional probabilities will not be of interest.

Now consider any $t < \alpha$, assume that event $\mathcal{A}$ holds. We then have that all $\hat{p}_j^Y > p_j^Y - t > 2\alpha - t > \alpha$. Combined with the fact that under event $\mathcal{A}$, $|\hat{p}_{ij} - p_{ij}| < t$ and $t \geq 0$, it is easy to check that

$$\begin{aligned}
\hat{p}_{i|j} - p_{i|j} &= \frac{\hat{p}_{ij}}{\hat{p}_j^Y} - \frac{p_{ij}}{p_j^Y} \\
&< \frac{p_{ij} + t}{p_j^Y - t} - \frac{p_{ij}}{p_j^Y} \\
&= \frac{p_{ij} p_j^Y + t p_j^Y - p_{ij} p_j^Y + t p_{ij}}{p_j^Y (p_j^Y - t)} \\
&< \frac{t p_j^Y + t p_{ij}}{p_j^Y \alpha} \\
&< \frac{2t}{\alpha},
\end{aligned}$$

where the last inequality follows since $p_{ij} < p_j^Y$ by definition. Similarly,

$$\begin{aligned}
p_{i|j} - \hat{p}_{i|j} &= \frac{p_{ij}}{p_j^Y} - \frac{\hat{p}_{ij}}{\hat{p}_j^Y} \\
&< \frac{p_{ij}}{p_j^Y} - \frac{p_{ij} - t}{p_j^Y + t} \\
&= \frac{p_{ij} p_j^Y + t p_{ij} - p_{ij} p_j^Y + t p_j^Y}{p_j^Y (p_j^Y + t)} \\
&< \frac{t p_j^Y + t p_{ij}}{p_j^Y 2\alpha} \\
&< \frac{t}{\alpha},
\end{aligned}$$

hence

$$|\hat{p}_{i|j} - p_{i|j}| < \frac{2t}{\alpha}.$$

Since by (37) the event $\mathcal{A}$ holds with probability at least $1 - 4 \exp(2 \log(n) - 2t^2 N)$, we have

$$\mathbb{P}\left( \max_{i,j \in 1,\dots,n} |\hat{p}_{i|j} - p_{i|j}| \geq \frac{2t}{\alpha} \right) \leq 4 \exp(2 \ln(n) - 2t^2 N). \tag{38}$$

The derivation of the bound for the conditional probability estimates in the other direction is similar and relies on the same event $\mathcal{A}$ holding. Hence the probability the bounds hold in both directions simultaneously remains $\mathbb{P}(\mathcal{A})$.

**Achieving error of** $\delta = 1/(n^2 \ln(n))$. Let $\alpha = \frac{\min\{\min_x p(x), \min_y p(y)\}}{2}$. Suppose we want $2t/\alpha = 1/(n^2 \ln(n))$. Then we need $t = 1/(2n^2\alpha^{-1}\ln(n))$. Note that $t < \alpha$ as required above. Suppose further that we want this to hold with probability at least $1 - 4/n$. By the above, we require

$$2\ln(n) - 2t^2 N < -\ln(n)$$

$$3\ln(n) < \frac{2N}{4n^4\alpha^{-2}\ln^2(n)}$$

$$6n^4\alpha^{-2}\ln^3(n) < N$$

Hence $N$ needs to be $\Omega(n^4\alpha^{-2}\ln^3(n))$. $\qquad\square$

# K   Proof of Theorem 3

From the equivalence between the minimum entropy coupling problem and the problem of finding the exogenous variable with minimum entropy, the output of $\mathcal{A}(\{\hat{p}(Y|X=x)\}_x)$ is the smallest entropy of any exogenous variable for the causal model $X \to Y$. Similarly, this claim holds for $\mathcal{A}(\{\hat{p}(X|Y=y)\}_y)$ as well. From Theorem 1, entropy in the direction $Y \to X$ scales with $n$ using $p(X|Y=y)$. From Theorem 6 of [6], it can be seen that the given sampling error can induce an entropy difference of at most $o(1)$ in the conditional entropies. Hence, even with noisy conditionals, $\max_y \hat{H}(X|Y=y)$ scales with $n$, implying that $\mathcal{A}(\{\hat{p}(X|Y=y)\}_y)$ scales with $n$. In the forward direction, the true exogenous variable provides a valid coupling under the true joint distribution without sampling noise. From Lemma 2, given $N$ samples, there exists a valid coupling in the forward direction that is constant entropy away from the true exogenous variable. Hence $\mathcal{A}(\{p(Y|X=x)\}_x)$ is constant. Since $\mathcal{A}(\{p(X|Y=y)\}_y)$ scales with $n$, the result follows.

# L   Proof of Theorem 4

We first show that the $H(X|Y=2)$ conditional entropy will have enough samples to be included in the criterion listed in Theorem 4. As $N = \Omega(n^2 \log(n))$, we have at least $c_1 n^2 \log(n)$ samples for $c_1 = \Theta(1)$. As shown in the proof of Theorem 1, $p(Y=2) = \Omega(\frac{1}{n}) \geq \frac{c_4}{n}$ where $c_4 = \Theta(1)$. Following a rejection sampling approach, we use Hoeffding's inequality to show that if $c_1 n^2 \log n$ samples are drawn from the joint distribution, then with probability $1 - o(1)$ we will successfully draw $\Omega(n \log(n))$ independent samples from the distribution $p(X|Y=2)$. Specifically, let $S_n$ denote the number of samples (out of $c_1 n^2 \log(n)$ total samples from the joint distribution) for which $Y = 2$, and $E_n = \mathbb{E}[S_n]$ denote the expected number of such samples. We have $E_n \geq (c_1 n^2 \log(n))(\frac{c_4}{n}) = c_1 c_4 n \log(n)$. Hence using Hoeffding's inequality,

$$P\left(S_n < \frac{c_1 c_4 n \log(n)}{2}\right) \leq P\left(|S_n - E_n| > \frac{c_1 c_4 n \log(n)}{2}\right) < 2e^{-\frac{2(c_1 c_4 n \log(n))^2}{c_1 n^2 \log(n)}} = 2e^{-2c_1 c_4^2 \log(n)} =$$

$o(1)$. Hence $S_n \geq \frac{c_1 c_4 n \log(n)}{2} \gg n$ with probability $1 - o(1)$. Thus the $\hat{H}(X|Y=2)$, which we use for identifiability, will have sufficient number of samples to be included in the criterion in Theorem 4.

We now show that each conditional entropy in the criterion in Theorem 4 will have error bounded by a constant with high probability. Immediately following from Corollary 1.12 of [30], for a distribution $D$ with support size $n$, $|H(D) - \hat{H}(D)| \leq 1$ with probability $1 - e^{-n^{c_2}}$ given a sample of size at least $\frac{c_3 n}{\log(n)}$ where $c_2, c_3 = \Theta(1)$. Since we only calculate conditional entropy estimates with $\geq n$ samples, the number of samples $n \gg \frac{c_3 n}{\log(n)}$ for all considered conditional entropies. Hence the total probability of any computed conditional entropy estimate being off by more than 1 is $\leq ne^{-n^{c_2}} = o(1)$ by the union bound. Since by the proof of Theorem 1 we know $\max_x H(X|Y=y) \leq c \ll \Omega(\log(\log(n))) \leq H(X|Y=2)$, it immediately follows that $\max_{x,\hat{p}(X=x)N \geq n} \hat{H}(Y|X=x) \leq c+1 \ll \Omega(\log(\log(n))) - 1 \leq \max_{y,\hat{p}(Y=y)N \geq n} \hat{H}(X|Y=y)$. $\qquad\square$

# M   Proof of Corollary 3

This generative model satisfies the assumptions of Theorem 3 following from the proof of Corollary 1. As such, under this generative model for sufficiently large $n$ and $N = \Omega(n^4 \alpha^{-2} \log^3(n))$ samples, $\mathcal{A}(\{\hat{p}(X|Y=y)\}_y) > \mathcal{A}(\{\hat{p}(Y|X=x)\}_x)$ with high probability.

We show a lower bound on $\alpha$ with high probability, under this generative model. As mentioned in the proof of Corollary 1, under this generative model, for any $i$, $P(x_i \leq z) = 1 - (1-z)^{n-1}$. We aim to show that with high probability, $x_i \geq \frac{1}{n^2 \log(n)}, \forall i \in [n]$ when $n$ is sufficiently large.

We lower bound the probability of this not happening as $(1 - (1 - \frac{1}{n^2 \log(n)})^{n-1})n$ by the union bound. Note that $\lim_{n \to \infty} \frac{(1-(1-\frac{1}{n^2 \log(n)})^{n-1})n}{1/\log(n)} = 1$.

Hence for sufficiently large $n$ the probability that there exists an $x_i < \frac{1}{n^2 \log(n)}$ is upper bounded by $\frac{2}{\log(n)}$. Thus, we have a high probability lower bound for $\alpha$. We substitute this for $\alpha$ in our lower bound for the number of required samples in the previous paragraph. This yields that under this generative model for sufficiently large $n$ and $N = \Omega(n^8 \log^5(n))$ samples, $\mathcal{A}(\{\hat{p}(X|Y=y)\}_y) > \mathcal{A}(\{\hat{p}(Y|X=x)\}_x)$ with high probability.

# N   Proof of Negative Association

**Lemma 9.** *Let $[x_i]_{i \in [n]}$ be a vector, uniformly randomly sampled from the probability simplex in $n$ dimensions. Then $[x_i]_{i \in [n]}$ is negatively associated.*

*Proof.* Let $x_i = \frac{z_i}{\sum_j z_j}$, where each $z_i$ is independent and identically distributed exponential random variable with mean 1, i.e. distributed as $\mathrm{Exp}(1)$. Then $[x_i]_i$ is a discrete probability distribution uniformly randomly chosen from the simplex in $n$ dimensions. We will show that $x_i$ are negatively associated. The following argument is provided by [23] as an answer on the online forum `https://mathoverflow.net/`, which we reproduce here for completeness.

Consider the following theorem:

**Theorem 5.** *[13] Let $z_1, z_2, \ldots, z_n$ be $n$ random variables with log-concave probability densities. Then $(z_1, z_2, \ldots, z_n)$ conditioned on $\sum_{i \in [n]} z_i$ are negatively associated.*

Note that exponential distribution is log-concave. Hence the theorem is applicable in our setting. Furthermore, the distribution induced on $(\frac{z_i}{\sum_{j \in [n]} z_j})_{i \in [n]}$ is identical to the distribution induced on $(z_1, z_2, \ldots, z_n)$ conditioned on $\sum_{i \in [n]} z_i = 1$. This concludes the proof.   $\square$

# O   Additional Experiments and Experimental Details

## O.1   Experimental Details

In this section, we provide the complete details of every experiment given in the main text, as well as provide additional results that we were not able to present in the main text due to space constraints.

**Sampling low-entropy exogenous variables:** We use Dirichlet distribution to sample the distribution for the exogenous variable from the probability simplex. Dirichlet has the parameter $\alpha$ which affects the entropy of the distribution obtained by sampling the corresponding Dirichlet distribution: Smaller $\alpha$ values lead to sampling distributions with smaller entropy. Suppose we want to sample distributions for $E$ such that $H(E) \leq \theta$. Since a good $\alpha$ value for this $\theta$ is not known a priori, we use the following adaptive sampling scheme: Suppose we want to sample $N$ distributions for $E$ such that $H(E) \leq \theta$. We initialize with $\alpha^{(0)} = 1$ and obtain $10N$ samples from Dirichlet with parameters $\alpha^{(0)}$. If there are at least $N$ samples out of $10N$ which has entropy less than $\theta$, we are done. If not, we set $\alpha^{(1)} = 0.5\alpha^{(0)}$ and iterate until for a particular $\alpha^{(i)}$ such that at least $N$ out of $10N$ samples satisfy the entropy condition.

Figure 6: Histogram of $H(\tilde{E})$ when $H(E) \approx 0.5 \log_2(n)$. Yellow line shows $x = 0.5 \log_2(n)$

**Details about Figure 2:** We set $E$ to have $mn$ number of states where $m, n$ are the number of states of $X$ and $Y$, respectively. It can be shown that this many number of states is sufficient to obtain any joint distribution. We uniformly randomly sample the function $f$ in the structural equation $Y = f(X, E)$. We also independently and uniformly randomly sample $p(X)$ from the simplex, i.e., we obtain samples from Dirichlet distribution with parameter $\alpha = 1$. For $m = n = 40$, we choose 20 values of $\theta$, i.e., entropy thresholds for the exogenous variable $E$, uniformly spaced in the range $[0, \log(m)]$. For $m \neq n$, we choose 10 $\theta$ values in the range $[0, \log(\max\{m, n\})]$.

When $m \neq n$, we use a mixture data as follows: We obtain 10000 samples from the graph $X \rightarrow Y$ and we obtain 10000 samples from $X \leftarrow Y$. We operate on this mixed data. This is done to reflect the fact that, there is no reason for the cause or the effect variable to have less or more number of states. Accuracy shown in the figures reflect the fraction of times each algorithm correctly identifies the true causal direction. Total entropy-based compares $H(X) + H(E)$ and $H(Y) + H(\tilde{E})$ where $E$ and $\tilde{E}$ are the outputs of the greedy minimum entropy coupling algorithm in the direction $X \rightarrow Y$ and $X \leftarrow Y$, respectively.

**Details about Figure 3:** We sample exogenous variable using the above adaptive sampling method so that, for each value of $n$, we have $H(E) \leq 0.8 \log(n)$. The other details are identical (e.g., 10000 samples for each configuration.) Due to the sampling method, we observe that most of the samples are very close to $H(E) \approx 0.8 \log(n)$. We then obtain the histogram plots for $H(\tilde{E})$, where $\tilde{E}$ is the output of the greedy minimum entropy coupling algorithm in the wrong direction. As observed, data fits well to a Gaussian and is highly concentrated around $0.854 \log(n)$.

**Details about Figure 5:** In this section, we introduce a latent confounder $L$. First, distribution of $L$ and distribution of $E$ are sampled independently. Then the distributions $p(X|l), p(Y|x, l, e)$ are sampled uniformly randomly from the simplex for every configuration of $x, l, e$. We use the adaptive sampling described above to sample $E$ such that $H(E) \leq 2$. Using the same sampling method, we sweep through different entropy thresholds for the latent confounder $L$ and sample such that $H(L) \leq \phi$ for $\phi \in \{0.5, 1, 1.5, 2, 2.5, 3\}$. The settings for $m, n$ and how data is mixed is identical to the procedure used to obtain Figure 2: When $m \neq n$, we use uniformly mixed data from $X \rightarrow Y$ and $X \leftarrow Y$. For each configuration, we obtain 1000 total number of samples and report the accuracy of the method to identify the true causal direction.

## O.2   Relaxing constant exogenous entropy assumption

As indicated in Section 4, we provide additional experiments for $\alpha = 0.2$ and $0.5$ in Figure 7 and Figure 6, respectively. As can be seen, for both $\alpha$ values, i.e., when $H(E) \leq \alpha \log(n)$, $H(\tilde{E})$ highly concentrates around $\beta \log(n)$ for some $\beta > \alpha$.

## O.3   Additional results on the finite sample regime

Figure 8 shows results on finite sample identifiability for the setting considered in the figure in the main text, except with smaller $H(E) \leq \ln(4)$.

Figure 7: Histogram of $H(\tilde{E})$ when $H(E) \approx 0.2 \log_2(n)$. Yellow line shows $x = 0.2 \log_2(n)$

(a) Identification via conditional entropies ($H(E) \leq \ln(4)$). (b) Identification via MEC algorithm ($H(E) \leq \ln(4)$). (c) Number of samples vs. support size of observed variables.

Figure 8: Finite sample identifiability of the causal direction via entropic causality. (a) Probability of correctly discovering the causal direction $X \rightarrow Y$ as a function of $n$ and number of samples $N$, using the conditional entropies as the test. (b) Probability of correctly discovering the causal direction $X \rightarrow Y$ using the greedy MEC algorithm to test the direction. (c) Samples $N$ required to reach 95% correct detection as a function of $n$, derived from the plots in Figure 8a and Figure 8b.

(a) Identification via conditional entropies ($H(E) = \ln(4)$). (b) Identification via MEC algorithm ($H(E) = \ln(4)$). (c) Number of samples vs. support size of observed variables ($H(E) = \ln(4)$).

Figure 9: Finite sample identifiability of the causal direction via entropic causality, where $p(x) \sim \text{Dir}(1)$ (uniform on the simplex). (a) Probability of correctly discovering the causal direction $X \rightarrow Y$ as a function of $n$ and number of samples $N$, using the conditional entropies as the test. (b) Probability of correctly discovering the causal direction $X \rightarrow Y$ as a function of $n$ and number of samples $N$, using the greedy MEC algorithm to test the direction. (c) Samples $N$ required to reach 98% correct detection as a function of $n$, derived from the plots in Figure 9a and Figure 9b.

Results for $p(X)$ drawn from $\text{Dir}(1)$ are shown Figure 9, as described in the main text. We find that the greedy MEC performance degrades to a level that is similar to the conditional entropy criterion. This might be explained by the fact that if $p(X|Y = y)$ are close to uniform, then the gap between $H(\tilde{E})$ and $H(X|Y = y)$ vanishes.

| | | | | | | | | |
|---|---|---|---|---|---|---|---|---|
| 5-state quantization | Threshold ($\times$ log support) | 0.6 | 0.7 | 0.8 | 0.85 | 0.9 | 1.0 | 1.2 |
| | # of pairs | 10 | 13 | 32 | 42 | 53 | 69 | 85 |
| | Accuracy (%) | 90.0 | 61.5 | 53.1 | 54.8 | 56.5 | 58.5 | 57.6 |
| 10-state quantization | Threshold ($\times$ log support) | 0.6 | 0.7 | 0.8 | 0.85 | 0.9 | 1.0 | 1.2 |
| | # of pairs | 8 | 12 | 23 | 39 | 49 | 71 | 85 |
| | Accuracy (%) | 87.5 | 66.7 | 60.9 | 53.8 | 51.0 | 52.1 | 57.6 |
| 20-state quantization | Threshold ($\times$ log support) | 0.6 | 0.7 | 0.8 | 0.85 | 0.9 | 1.0 | 1.2 |
| | # of pairs | 5 | 10 | 15 | 31 | 54 | 78 | 85 |
| | Accuracy (%) | 60.0 | 70.0 | 73.3 | 54.8 | 48.1 | 48.7 | 55.3 |

Table 3: Performance on Tübingen causal pairs with low exogenous entropy in at least one direction. Chosen based on majority voting on 5 random quantizations.

## O.4  Additional Tuebingen Experiments

In this section, we perform additional experiments to evaluate the stability of the method to choice of quantization on the Tuebingen dataset. Specifically, to quantize $[a, b]$ into $n$ intervals, we perturb each quantization point $\{a + \frac{(b-a)i}{n}\}_i$ with a uniform noise in $[-\frac{(b-a)}{8n}, \frac{(b-a)}{8n}]$. For every pair, this is done 5 times independently and the majority decision is taken. The results, which show similar performance to Table 1 are shown in Table 3, demonstrating a degree of stability to choice of quantization. We observe that perturbed quantization demonstrates better performance for $20-$state quantization, whereas it shows somewhat worse performance for the 5 and $10-$state quantizations. This indicates that more research is needed to determine the optimal quantization for a given dataset.

## Footnotes

[2]This relabeling of $X, E$ is without loss of generality since realization of $f$ is symmetric across rows and columns.

[3]The probabilities we sum correspond to disjoint events, hence the total probability cannot exceed $1$.

[4] Uniformity follows from uniformity of $f$.