[Reviews · NeurIPS 2020]

Review 1

Summary and Contributions: The paper describes an approach that renders the cause-effect problem solvable by preferring structural equation models with low entropy noise variables

Strengths: Provided that one is willing to accept the assumptions, this paper is a nicely written and a technically sound contribution. It provides a novel approach to the cause-effect problem that is worth discussing.

Weaknesses: What I miss is a more critical conceptual discussion of the assumptions. Although it seems appealing to prefer low entropy for the noise distributions, this assumption raises lots of questions. Janzing, [*] for instance, challenged the common idea that the direction that admits a deterministic model is always the causal one. It states that independence of mechanisms (although hard to formalise) is the more reliable concept. One may consider Janzing's counter example contrived and hand-tuned, but it is at least worth discussing. Taking it for granted that the deterministic direction is the causal one sounds a bit too simple. The assumption of low entropy noise seems worth discussing for several reasons: If one assumes, on the other hand, that the cause itself has large entropy, I don't see a systematics behind the approach, given that the cause variable is its own noise variable in a typical structural equation. Shouldn't one assume this variable to also have small entropy? [*]  D. Janzing. The cause-effect problem. The cause-effect problem: motivation, ideas, and popular misconceptions https://docs.google.com/a/chalearn.org/viewer?a=v&pid=sites&srcid=Y2hhbGVhcm4ub3JnfGNhdXNhbGl0eXxneDphZjdhZWE1MTU2MGVlNDQ  misconceptions.

Correctness: The paper seems sound from the purely formal point if view, assumptions and results are clearly stated. All my concerns are conceptual. Apart from the above, I am also struggling with probabilistic statements that rely on functions drawn randomly from the set of all functions. Nature has more structure, assuming that it draws uniformly from the set of all functions would render all learning problems unsolvable.

Clarity: The paper is well-written.

Relation to Prior Work: The background of the paper I sufficiently described. Not mentioning other information theoretic causal inference may be seen as a moderate omission though. For the special case of additive note models, http://proceedings.mlr.press/v32/kpotufe14.html describes an information theoretic approach to inferring directions which accounts for the entropy of the hypothetical cause and the entropy of the noise. This should definitely be mentioned.

Reproducibility: Yes

Additional Feedback: The broader impact remarks do not sound convincing to me since drug testing problems usually raise the question of telling confounding from cause-effect relations rather than inferring the causal direction. Some related work that comes to my mind since it is also an information theoretic approach to the cause-effect problem is information geometric causal inference (IGCI): https://www.sciencedirect.com/science/article/pii/S0004370212000045 https://link.springer.com/chapter/10.1007/978-3-319-21852-6_18 IGCI is sort of complementary to the present work, since it assumes determinism in both directions. I therefore don't consider not discussing it as a major thing. But section 3 in the second reference sounds related to the present approach. I raised my score since I trust the authors that they include the following for their final version: - mention that focusing on complexity (entropy) of function may not be sufficient - mention that preferring deterministic direction as the right one may fail - mention that applying the idea to the structural equation model of the cause would result in low entropy cause, briefly discuss whether this fact renders generalisation to more than 2 variables problematic


Review 2

Summary and Contributions: The paper investigates identifying a causal direction between two categorical variables from observational data. The authors offer the conditions under which the causal direction between the two variables can be determined with high probability by estimating entropy of the variables and noises. Further, the paper provides the sample complexity for a finite sample case. The authors empirically validated their theoretical results by conducting extensive experiments.

Strengths: Concrete theoretical results both on infinite/finite sample regimes. Empirical evaluation is well-executed to provide how the violations of the assumptions affect the results. The results are novel. Given the interest within NeurIPS community on causality in general, the results in this paper will attract many attendees.

Weaknesses: The assumption that the two categorical variables are not confounded is a bit stringent but this is what this paper is about to focus on. Further, it does not investigate whether the results can be generalizable to the case of a simple graph. For example, given a true graph A -> B -> C -> D, can we use the results to determine the direction between B and D with A and C unobserved? (where we usually draw a graph B -> D (as known as a latent projection) Hence, the significance is relatively moderate considering real-world use-cases.

Correctness: The result seems correct although I was not able to rigorously check the correctness of the claims in the paper (mostly pointed to the supplementary material).

Clarity: The paper is clearly written providing proper explanations after each conjecture/theorem.

Relation to Prior Work: The prior work by Kocaoglu provides a conjecture about the identifiability of causal direction and this paper modifies the conjecture, and provides under which conditions the causal direction can be identified with high probability.

Reproducibility: Yes

Additional Feedback: I have read the authors' feedback. ---------- Thanks for the great work. I have a few questions and comments. It is unclear to me whether the use of the term “identifiability” in this setting is appropriate. Identifiability is usually used in a stricter sense in causal inference literature, e.g., whether you can “pinpoint” the causal effect of interest. The paper discusses whether the causal direction can be “accurately” predictable. Further, the high probability guarantees in this paper are up to the assumption that P(X), P(E), and f are sampled uniformly randomly, which provides the prior distribution about the underlying world. Then, the result is comparing the posterior distributions for H(X)+H(E) and H(Y)+H(tilde E). I am wondering whether the word “identifiability” can be replaced to other terms. Further, whether the term “Bayesian” can be specified either in the title. For the quantization approach (the last experiment), can you make use of an ensemble approach? For example, there are many ways to quantize data into e.g., 5 random states. Then, you can employ a majority-vote to determine the direction. This experiment seems useful to investigate whether the results in this paper can be generally applicable to the continuous setting. I am also wondering whether you can employ bootstrapping to report confidence intervals for the accuracies in Table 1. Additionally, is it possible to employ/develop a supervised quantization method enforcing minimum entropy? Finally, is it possible to output the probability of the causal direction? minor Figure 1 “distribution of …” → “domain of …“ Figure 3. subgraphs need to be aligned (font size, etc) Figure 4 (a) (b) the x-axes only show 10^4. Is it because it is between 2000 ~ 60000 in log scale? (need some other labels to parse the graphs better) Tuebingen → T\”ubingen footnote 1 seems unnecessary. I hope the conclusion is more informative. check the parentheses of the equation at Line 571 in the supplemental material. (closing & opening parentheses do not match.) Line 183 having a finite number of I am unsure whether the topics you mentioned in the broad impact section are relevant to the results presented in this paper.


Review 3

Summary and Contributions: This paper builds on a line of work in causal inference that uses the entropy of exogenous noise to determine the causal direction. The main contributions include proving a modified conjecture from [11] about identifiability using Shannon entropy and analyzing the sample complexity.

Strengths: While [11] did include identifiability results for Renyi entropy with a=0 (cardinality), they only had a conjecture for identifiability for Shannon entropy. This work proves a modified form of that conjecture. This work also rigorously analyzes sample complexity. The authors comment or empirically evaluate several potential concerns arising from technical restrictions from the proofs. There are a number of experiments included. For the specific topic of entropic causal inference, and (to a lesser extent) the broader field of causal inference, I think this is a significant contribution.

Weaknesses: In terms of the work done, namely the analysis, empirical evaluations, and writing, I would say there are no major weaknesses. I am impressed with the results the authors obtained and overall find the paper well-written. Though I do have some concerns about the broader significance, part of which is inherited from the interesting but perhaps inherently challenging problem of entropic causal inference [11]. -- [11] and this paper only consider the case of two variables -- identifying the minimum entropy exogenous variable with just two observed variables is NP hard (though [11] has a greedy algorithm and that was later improved upon) -- in this paper, the analysis requires the entropy of the exogenous variable to be constant with respect to the cardinality and entropy of the observed variables. (The authors provide experimental evidence suggesting that logarithmic growth may be fine.) -- the results in this paper are couched in the setting of the cardinality $n$ of the observed variables growing, which is a little strange.

Correctness: I did not go through the supplementary materials, which included proofs and details about empirical evaluations. But in my reading of the main submission and skimming the supplementary material, no results stood out as being wrong or suspicious. The only concern about correctness I have is the contribution claim "We prove the first identifiability result for the entropic causal inference framework" [11] includes an identifiability result for Renyi entropy with a=0 (measuring cardinality). This is the first work I am aware of that had identifiability for Shannon entropy ([11] only had a conjecture).

Clarity: Overall I found the paper fairly well-written. Some of the results are quite technical, leading to many parameters and formulas, but the authors take effort to explain them.

Relation to Prior Work: Yes, to my knowledge [11] is the most important related work, and with the exception of the claim about being the first to have identifiability results (described above), the discussion is clear. The authors go into more detail on related work in the supplementary material; entropic causal inference itself can be seen as a generalization of the additive noise model. The authors include references to that literature in the supplementary material.

Reproducibility: Yes

Additional Feedback: minor comment -- in Theorem 3 and Corollary 3, \mathcal{A} is said to be an algorithm that solves the minimum entropy coupling problem, so the output would be a distribution, but the notation of lines 209-210 and line 217 suggests the output being the entropy of the exogeneous variable.


Review 4

Summary and Contributions: This paper shows that for almost all causal models where the exogenous variable has entropy that does not scale with the number of states of the observed variables, the causal direction is identifiable from observational data. In addition, the paper demonstrates that this result could provide important identifiability guarantees for existing algorithms. Furthermore, the paper provides experiment results on (1) the robustness of the method after relaxing some assumptions, and (2) empirical number of observational samples needed for causal identification (3) application to real-world dataset.

Strengths: The paper is very well-written, providing solid overview of the field and intuitive explanations for the problem and the results. Moreover, the paper addresses an important problem in the field, which could have strong impact on how causality relations could be learned.

Weaknesses: In the experiment section, it would be great to see more details on how the synthesized data is generated, potentially in supplementary.

Correctness: N/A (please see section for "confidential comments for the Area Chair")

Clarity: Yes, it paper is very well written.

Relation to Prior Work: Yes.

Reproducibility: No

Additional Feedback: For reproducibility, it would be great to see more details on how the synthesized data is generated, potentially in supplementary.


Review 5

Summary and Contributions: This work is focused on finding the causal relation between two categorical variables. The idea is that if the system is close to a deterministic system, in the sense that the entropy of the exogenous variable in the causal direction does not scale with the alphabet size of the variables and the alphabet size is large enough, then under some extra assumptions, the entropy of the exogenous variable in the non-causal direction is larger than that of the causal direction. Therefore, the causal direction is identifiable. A finite-sample analysis and bounds on the number of samples needed for identification is also provided.

Strengths: The authors study the important problem of learning the causal direction between two variables, which has applications in many fields and is of highly relevance to the ML society. They aim to address this problem using information theoretic machinery which has received less attention and can motivate future work.

Weaknesses: The motivation of the work originates from an observation regarding deterministic system: If the system is deterministic, that is the effect variable is a deterministic function of the cause variable, in the non-causal direction, most of the functions will be one-to-many, and hence the causal direction is easily identifiable. The authors then aim to extend this idea to non-deterministic systems. To do so, (among several other assumptions) it is assumed that the entropy of the exogenous variable corresponding to the effect variable is bounded in the true causal direction. Therefore, if the alphabet size of the variables is large enough, the entropy in the non-causal direction will be larger than that of the causal direction and hence, the causal direction is identifiable. Therefore, the result easily follows by extending the idea of deterministic systems. Here, a clear concern is that we can never know if the alphabet size is large enough to be able to estimate the system as a deterministic system. This assumption is fundamentally different from other existing assumptions in the literature that, e.g., assume additivity of the noise. It seems that the assumption in this work forces the model to be more deterministic in one direction and then uses a measure of randomness for the identification. There are three main assumptions in this work: Causal sufficiency, Assumption 1 and forcing the entropy of exogenous variable in the causal direction to be limited. The assumptions have been stated without any clear description of their implications. It should be clarified what restrictions and limitations do these assumptions force on the data and data generating process. In the proposed approach, the entropy of the exogenous variable in one direction should be compared to the entropy of the other direction. What should be done if more than one model can be fit to the data? It seems that in general, one should be able to fit more than one function f and exogenous variable E to the data. Same for function g and variable E tilde. Unfortunately the method is not compared with other existing methods in the literature such as methods for additive noise (Hoyer et al.), LiNGAM (Shimizu et al.), information-geometric approach (Janzing et al.), etc. In many of the result statements, the authors have only vaguely said "with high probability". These statements should be stated more rigorously.

Correctness: Yes.

Clarity: Yes.

Relation to Prior Work: A comparison with other methods for finding causal relation between two variables is missing.

Reproducibility: Yes

Additional Feedback: I thank the authors for their responses. My score remains unchanged.

[Author Response · NeurIPS 2020]

We sincerely thank the reviewers for their valuable feedback. We are glad to see that the reception of our paper has been mostly positive. R2 requested a conceptual discussion of the assumptions. R7 had clarification/improvement questions. We address these questions and the others below. All minor comments will be implemented in the camera-ready.

**R2:** *Philosophy of Low-Entropy Assumption.* There are two philosophical interpretations. First one is the statement that the true causal model has a small amount of randomness. This is the interpretation taken in [11]. Note that this requires a comparison between $H(X) + H(E)$ vs. $H(Y) + H(\tilde{E})$. Even though this incorporates $H(X)$ as you point out, it does not require $H(X)$ to be small, but the total entropy to be smaller. The second interpretation is that "the **additional** unobserved randomness is small", which can be seen as relaxing the determinism assumption. Our objective is to quantify how much we can relax this assumption and still retain identifiability. A different interpretation is that entropy of the model can be seen as a way to approximate its Kolmogorov complexity. Kolmogorov complexity of a causal model has been proposed by Janzing et al. in [8] as a way to identify the causal direction. We are going to point out this connection and hope to make this a more formal connection in the future. Thank you for pointing out to Janzing et al. We will discuss this counterexample in relation with our assumptions.

*Nature has structure.* We agree that if we know the structure, this could help. Assuming uniform distribution over the function space is our way of measuring how often the proposed method may fail, given that we do not know nature's structure. This assumption can be relaxed in specific ways, for example when there is one state $y$ whose inverse image is largely supported, this is sufficient. We see this as an indication that uniform assumption is not necessary. We will flesh out these alternatives to uniform sampling of $f$ in camera-ready.

*Related work.* We will add all the citations along with a detailed discussion on how they compare to our approach.

**R3.** *No confounder assumption.* We have provided experimental results to illustrate that the proposed method is robust to light confounding. Please see Section 4 lines 277-287.

*What can be said about line graphs?* Thank you for pointing this out. As long as exogenous entropy for each variable on the line graph is a constant and there are not too many such variables, our machinery can be applied. $B$ and $D$ in your example is a valid case. This is due to the fact that we can write $B \rightarrow D$ with exogenous entropy of at most $H(E_C) + H(E_D)$. We will explain this and other relevant settings in camera-ready.

*Identifiability* While used in multiple ways in literature, the term identifiability within the causal discovery settings typically refers to identifying the causal direction/graph. We will point to the related work that employs the same usage.

*Bayesian interpretation.* We agree that one can interpret the assumptions as being Bayesian. However, the posterior distributions for these quantities are very hard to compute. Therefore, we refrained from this terminology. Instead, we put a measure on the model space to be able to quantify what fraction of models can be identified using the proposed framework. If a prior on the function space is known, this can be incorporated in our proof to understand whether identifiability persists. Without any knowledge, we believe uniform prior is suitable.

*Comments on quantization.* We will try the ensemble approach and report in camera-ready. We chose to use the whole dataset rather than bootstrap, since the number of samples is small ( 100). Thank you for the other research directions.

**R4:** Thank you for your feedback. We will clarify that Renyi 0 case is resolved in [11].

**R6:** Please see Section O (line 820) of Appendix for the details about the synthetic data generation. We will move these details to the main text, making use of the extra page in camera-ready. Thank you.

**R7:** *Implications of the assumption* We provided several experimental results to assess the effect of our assumptions. In Section 4 lines $237 - 253$ we experimentally evaluate the implications of the low exogenous entropy assumption. If $H(E)$ is too small, this implies $H(X) > H(Y)$. If it is large enough, this implies $H(X) < H(Y)$. This creates three different regimes. We show in Fig. 2 that our method almost always identifies the causal direction in all three regimes.

*What if more than one model can be fit to the data?* We aim to find the model with the minimum exogenous entropy in both directions. Even if there are multiple such models, the minimum entropy will be unique. Also note that our bounds hold for *any E for which there exists an f s.t.* $Y = f(X, E)$, thus providing us a lower bound to the minimum entropy.

*Comparing with existing methods.* The performances of these methods on Tübingen are available in the literature (see [15] by Mooij et al.). We will include the accuracies reported by them. As far as we are aware, ANM provides around $64\%$ accuracy. Information-geometric (IGCI) approach performs worse than ANMs. A nonlinear extension of LiNGAM gives $62 - 69\%$ as reported in Hyvarinen et al. "Pairwise Likelihood Ratios for Est. of Non-Gaussian SEMs". Also note that, except IGCI, all of these methods require ordinal variables, whereas we can handle categorical data.

*We never know alphabet size is large.* In practice, we fit the simplest model in both directions and pick the one with the smaller entropy, regardless of the alphabet size. In our experiments, we set a threshold $t < 1$ and make a decision only if $H(E) \leq t \log(n)$ or $H(\tilde{E}) \leq t \log(m)$. Please see Table 1. Thank you for your feedback.

[Meta-Review · NeurIPS 2020]

In this paper, the authors continue a previous line of work (initiated in the cited reference [11]) and prove some interesting results on the finite-sample identifiability of causal pairs under an "entropic identifiability" assumption. In the end, there was substantial discussion and reviewers were split on this paper, raising several crucial issues that need to be carefully addressed by the authors. Please pay close attention in particular to the comments by R2, R7. Below I outline the major issues that must be addressed in the camera ready: - Please add a critical discussion and comparison with existing work on identifiability in causal models. Given the prior work [11] and the lack of substantial discussion or comparisons with existing methods for inferring causal pairs, the significance of this work is left unclear. There is already substantial literature on identifiability in causal models and causal pairs in particular, and a proper discussion to situate this work within this prior work is missing. - Add detailed comparisons with existing methods. The authors mention that such comparisons are available by manual inspection of existing papers, however, this must be collected in one place and critically discussed in the paper. (As proposed by the authors in their response, this should not require any additional experiments.) - A critical discussion of Theorem 1: The authors only prove a high probability result under strong assumptions, which is strictly weaker than even very weak notions of identifiability such as generic identifiability. This is an unusual approach to identifiability, and this warrants additional discussion. See the comments from R3. - In several places, technical jargon is used without clarification (e.g. "high probability", "O(1)", etc.). For example, in Conjecture 1, O(1) is used even though this is not an asymptotic statement. Similarly, "with high probability" is used without clarifying what the authors mean by this (again, Conjecture 1 is not an asymptotic statement). These constants and probabilities as well as their dependencies (e.g. what is going to infinity?) should be made explicit. Ultimately, the substantive ideas seem to have appeared previously, and the main contribution here is a detailed technical analysis that rests on very strong assumptions. Additional discussion is needed to properly situate this work and identify its limitations.